# QuanTA: Efficient High-Rank Fine-Tuning of LLMs with Quantum-Informed Tensor Adaptation

**Zhuo Chen**[12]    **Rumen Dangovski**[13]    **Charlotte Loh**[13]

**Owen Dugan**[12]    **Di Luo**[124*]    **Marin Soljačić**[12]

[1]NSF AI Institute for Artificial Intelligence and Fundamental Interactions
[2]Department of Physics, Massachusetts Institute of Technology
[3]Department of EECS, Massachusetts Institute of Technology
[4]Department of Physics, Harvard University

{chenzhuo,rumenrd,cloh,odugan,diluo,soljacic}@mit.edu

## Abstract

We propose **Quan**tum-informed **T**ensor **A**daptation (**QuanTA**), a novel, easy-to-implement, fine-tuning method with no inference overhead for large-scale pre-trained language models. By leveraging quantum-inspired methods derived from quantum circuit structures, QuanTA enables efficient *high-rank* fine-tuning, surpassing the limitations of Low-Rank Adaptation (LoRA)—low-rank approximation may fail for complicated downstream tasks. Our approach is theoretically supported by the universality theorem and the rank representation theorem to achieve efficient high-rank adaptations. Experiments demonstrate that QuanTA significantly enhances commonsense reasoning, arithmetic reasoning, and scalability compared to traditional methods. Furthermore, QuanTA shows superior performance with fewer trainable parameters compared to other approaches and can be designed to integrate with existing fine-tuning algorithms for further improvement, providing a scalable and efficient solution for fine-tuning large language models and advancing state-of-the-art in natural language processing.

## 1 Introduction

Pre-trained large language models (LLMs) have revolutionized natural language processing (NLP) by achieving state-of-the-art performance across various tasks [1, 2]. Traditionally, these models are adapted to specific downstream applications via full fine-tuning, where all model parameters are retrained. However, as model sizes increase, the computational cost and memory requirements for full fine-tuning become prohibitive, especially with models like GPT-3 [3] with 175 billion parameters, Mixtral [4] with $8 \times 22$ billion parameters, and more recently the LLaMA series [5–7], containing soon up to 400 billion parameters [8]. These constraints have spurred the development of parameter-efficient fine-tuning (PEFT) methods, which aim to adapt LLMs by updating only a small subset of parameters, thereby reducing resource demands [9, 10].

Among PEFT methods, Low-Rank Adaptation (LoRA) [10] has gained prominence due to its simplicity and effectiveness. LoRA fine-tunes LLMs by introducing low-rank matrices into the pre-trained model's weight updates, pragmatically reducing the number of trainable parameters while maintaining performance close to full fine-tuning in many tasks. However, LoRA's reliance on low-rank approximations can sometimes lead to a performance gap compared to full fine-tuning, particularly for complex tasks, as it may not capture all necessary task-specific adaptations [11].

38th Conference on Neural Information Processing Systems (NeurIPS 2024).

*Corresponding author.

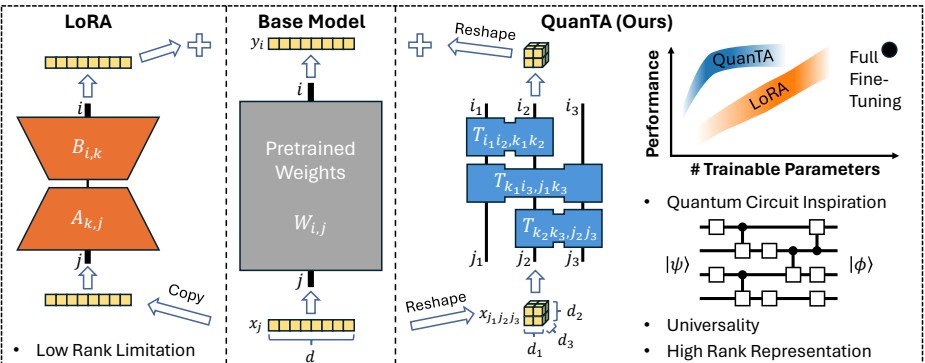

Figure 1: Conceptual comparison of QuanTA and LoRA methods. LoRA parameterizes the weight matrix update as a outer product of two low-rank matrices, limiting its capacity. QuanTA, inspired by quantum circuits, uses tensors that operate on specific axes of the (reshaped) input, enabling high-rank parameterization. Supported by the universality theorem and rank representation theorem, QuanTA can represent arbitrary matrices effectively, allowing it to achieve performance comparable to or sometimes even better than full fine-tuning, with only a fraction of the parameters. Note: the performance graph is a conceptual illustration.

Recently, there have been many attempts to generalize LoRA using tensor-based methods [12, 13]. However, these approaches primarily focus on reducing the number of trainable parameters within the low-rank framework yet they continue to face the same limitations of restricted representation. In Quantum mechanics, quantum circuit provides a natural realization of unitary matrix which is full rank, motivating us to develop new schemes for high-rank fine-tuning.

Inspired by these advancements, we propose **Quan**tum-informed **T**ensor **A**daptation (**QuanTA**) * a novel, easy-to-implement, fine-tuning method with no inference overhead inspired by quantum circuits (Fig. 1). QuanTA enables efficient high-rank adaptations by utilizing tensor operations analogous to those in quantum circuits, addressing the limitations inherent in low-rank methods like LoRA.

In summary, our contributions are as follows:

1. We introduce QuanTA, a novel, easy-to-implement, PEFT method with no inference overhead inspired by quantum circuits, enabling efficient high-rank fine-tuning without additional inference latency and offering the potential for integration with other existing PEFT methods for further enhancement.

2. We present the universality theorem and the rank representation theorem, theoretically proving that QuanTA can efficiently parameterize high-rank matrices, overcoming the limitations of low-rank methods.

3. We validate QuanTA's performance through extensive experiments, demonstrating significant improvements in various reasoning tasks and efficiency compared to traditional methods.

## 2 Related Works

**Parameter-Efficient Fine-Tuning (PEFT)** methods aim to address the computational burdens associated with fine-tuning large-scale models by adjusting a relatively small fraction of the total parameters to fit a specific downstream task. Roughly speaking, there are three existing categories of PEFT methods:

1. **Adapter-based methods.** These methods introduce additional trainable modules into the structure of a pre-trained, otherwise frozen, model. These modules can be integrated in various ways: series adapters are interposed between existing layers like attention or MLP components [9, 14–16], while parallel adapters coexist alongside these components [17]. In general, these methods tend to increase the inference load due to the extra components that are not readily integrated into the original model weights.

---

*https://github.com/quanta-fine-tuning/quanta

2. **Prompt/Prefix-based methods.** These methods employ additional prompts or soft tokens at the beginning of the input sequence, focusing fine-tuning efforts on these newly introduced vector embeddings while maintaining the original model weights static [18, 19]. However, this approach can suffer from suboptimal performance and increased inference times. In addition, the soft tokens take up space of real tokens and therefore reduce the effective context size available for the model.

3. **Reparameterization-based methods.** These methods modify the existing weights with some parameter-efficient parameterization during the fine-tuning phase. Among these methods, Low-Rank Adaptation (LoRA) [10] and its variants, such as DoRA [20] and VeRA [21], are particularly noteworthy for their widespread adoption and robust performance across various tasks. In addition to LoRA, many other PEFT methods also belong to this category, including more sophisticated approaches such as Hadamard [22], Kronecker product [23] reparameterizations as well as many other methods [24–27]. Crucially, methods in this category do not impose additional inference burdens after fine-tuning as the modified weights can be merged into the pre-trained model weights prior to deployment.

Besides these three categories, there are additional PEFT methods such as LoTA [12], where tensor decompositions are performed across multiple weights, LoRETTA [13], which uses tensor train decomposition for each weight matrix and has both adapter-based and reparameterization-based variants, MPO-based fine-tuning [28], and very recently LISA [29], ReFT [30] and MoRA [31].

**Physics-inspired machine learning** In parallel, there have been various attempts to integrate physics-based priors into machine learning for many years. Symmetries and physics structure have been incorporated into the neural networks architecture and training in various applications to achieve notable performance [32–39]. Various classical and quantum physics processes have been utilized to design new neural networks [40, 41] and generative models [42–48].

## 3 Motivation: Low Rank is not Always Sufficient

LoRA operates under the hypothesis that parameter updates during fine-tuning exhibit a low "intrinsic rank." For a pretrained weight matrix $W_0 \in \mathbb{R}^{d \times k}$, LoRA parameterizes the weight update as $W' = W_0 + \Delta W = W_0 + BA$, where $A \in \mathbb{R}^{r \times k}$ and $B \in \mathbb{R}^{d \times r}$ are low-rank matrices. In this configuration, only $A$ and $B$ are trainable, while $W_0$ remains fixed. Consequently, the rank of the weight update $\Delta W$ is limited to $r$.

Although the original LoRA paper shows empirical evidence to support the low-rank hypothesis, recently it has been found that this hypothesis may still fail for more complex tasks, especially for those that significantly differ from the pre-training dataset, leading to suboptimal performance [11, 31]. To assess the general applicability of the low-rank hypothesis, we examine two datasets of varying difficulties: the RTE dataset [49], a classification task where the model is tasked to verify the correctness of statements, and the DROP dataset [50], a generation task where the model performs discrete reasoning over paragraphs. We posit that the RTE dataset is simpler, thus more likely to conform to the low-rank hypothesis, whereas the DROP dataset presents a greater challenge.

As shown in Table 1, the LLaMA2-7B model [6] in general can achieve a better score on the RTE dataset than the DROP dataset. In addition, as we increase the rank from 64 to 128, LoRA's performance on the RTE dataset remains the same, consistent with

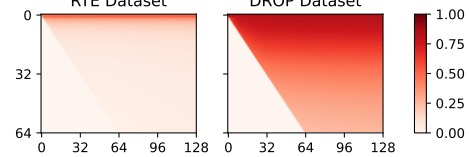

Figure 2: Subspace similarities between two LoRA experiments of different ranks (64 and 128) for two datasets. Each point $(i, j)$ represents the subspace similarity between the first $i$ right singular vectors of the $r = 64$ experiment, and the first $j$ right singular vectors of the $r = 128$ experiment. Only points for $i \leq j$ are plotted. DROP dataset has a significantly high "intrinsic rank" than RTE dataset.

| Model | Accuracy/$F_1$-Score (↑) | |
| --- | --- | --- |
| | **RTE** | **DROP** |
| LLaMA2$_{7B}$ Base | 61.0 | 19.8 |
| LLaMA2$_{7B}$ LoRA$_{r=64}$ | 86.0 | 55.2 |
| LLaMA2$_{7B}$ LoRA$_{r=128}$ | 85.8 | 56.2 |

Table 1: Performance of base and LoRA fine-tuned LLaMA2-7B on RTE [49] and DROP [50] datasets. We use accuracy and $F_1$-score as the metrics for them respectively.

the low-rank hypothesis, while the performance on the DROP dataset improves, suggesting the DROP dataset may require a higher "intrinsic rank."

To further measure the "intrinsic rank" of weight updates for these datasets, we follow the methodology outlined in [10] and compare the subspace spanned by the right singular vectors of the resulting weight updates between the $r = 64$ and $r = 128$ experiments. Figure 2 shows the subspace similarities between the query weight updates of the two ranks at layer 16 for both datasets. In the figure, each point $(i, j)$ represents the subspace similarity between the first $i$ singular vectors of the $r = 64$ experiment and the first $j$ singular vectors of the $r = 128$ experiment. A subspace similarity close to 1 indicates significant overlap, suggesting that the subspace is crucial for fine-tuning, while a similarity close to 0 suggests orthogonality, implying that the vectors represent noise. For the RTE dataset, subspace similarity is large only for very small $i$ values, and quickly decays to 0 for larger $i$, indicating that fine-tuning on the RTE dataset has a low "intrinsic rank." Conversely, for the DROP dataset, subspace similarity remains large across all 64 singular vectors, demonstrating a high "intrinsic rank." Additional details of subspace similarity and addition data are provide in Appendix A

These findings demonstrate the necessity of high-rank fine-tuning in complex tasks, challenging the effectiveness of LoRA. This naturally prompts the following question: *How can we design efficient methods to facilitate high-rank updates during fine-tuning?*

## 4 Preliminary: Quantum Circuit

The behavior of quantum mechanical systems, especially those involving particles with discrete degrees of freedom, is well described by matrix theory. Quantum circuits naturally realize unitary matrices whose sizes grow exponentially with the number of particles, providing a potent framework for high-rank representation. Here, we review some fundamental concepts of quantum states and quantum circuits to motivate our approach.

**Quantum state and vector representation.** An $N$-qubit quantum state $|\psi\rangle = \sum_i \psi_i |i\rangle \in \mathbb{C}^{2^N}$ is a $2^N$-dimensional complex-valued vector in Hilbert space, with $\psi_i$ the components and $|i\rangle$ the basis vectors (similar to $\mathbf{e}_i$ in vector notation). Since quantum states typically consist of qubits with local dimensions of 2, it is instructive to view the quantum state as a multi-dimensional tensor with different indices labeling different qubits: $|\psi\rangle = \psi_{i_1,i_2,\ldots,i_N} |i_1, i_2, \ldots, i_N\rangle$, where $i_1, i_2, \ldots, i_N$ is the binary representation of $i$. This can be equivalently viewed as reshaping the quantum state from a vector in $\mathbb{C}^{2^N}$ to a tensor in $\mathbb{C}^{2 \times 2 \times \cdots \times 2}$.

**Quantum circuit and matrix representation.** A quantum circuit is a unitary matrix $\mathcal{U} \in \mathbb{U}(2^N) \subset \mathbb{C}^{2^N \times 2^N}$ that transforms one quantum state into another: $|\phi\rangle = \mathcal{U} |\psi\rangle$. These circuits are constructed from smaller unitary matrices known as quantum "gates," which operate on one or two qubits. A one-qubit gate is a unitary matrix $U^{(1)} \in \mathbb{U}(2^1)$, while a two-qubit gate is a unitary matrix $U^{(2)} \in \mathbb{U}(2^2)^\dagger$. These gates are applied to specific qubits as follows:

Figure 3: Any unitary matrix can be decomposed into a quantum circuit using one- and two-qubit gates.

$$U^{(1)} |\psi\rangle = \sum_{j_n} U^{(1)}_{i_n;j_n} \psi_{i_1,i_2,\ldots,j_n,\ldots,i_N} |i_1, i_2, \ldots, i_N\rangle \tag{1}$$

for a one-qubit gate applied to qubit $n$, and

$$U^{(2)} |\psi\rangle = \sum_{j_m,j_n} U^{(2)}_{i_m,i_n;j_m,j_n} \psi_{i_1,i_2,\ldots,j_m,\ldots,j_n,\ldots,i_N} |i_1, i_2, \ldots, i_N\rangle \tag{2}$$

for a two-qubit gate applied to qubits $m$ and $n$. (Note that $m$ and $n$ do not need to be consecutive qubits.)

A quantum circuit comprises a series of these one- and two-qubit gates $\{U^{(\alpha)}\}$ applied sequentially to the quantum state:

$$\mathcal{U} |\psi\rangle = \prod_\alpha U^{(\alpha)} |\psi\rangle. \tag{3}$$

---

$^\dagger$Typically, quantum circuits and quantum gates are considered within the group $\mathbb{SU}(2^N)$. However, the groups $\mathbb{SU}(2^N)$ and $\mathbb{U}(2^N)$ differ only by a $\mathbb{U}(1)$ factor, which does not affect the results presented in this paper.

Since quantum circuits are unitary, they inherently represent full-rank matrices in finite-dimensional systems.

**Universality of quantum circuit.** Similar to the universal approximation theorem for neural networks, it has been established that any quantum circuit on $N$ qubits can be decomposed into a quantum circuit using only one- and two-qubit gates [51–53], as shown in Figure 3. This is particularly relevant for reparameterization-based fine-tuning methods, where we aim to parameterize a matrix matching the shape of the base model's weight matrix using a small number of parameters.

## 5 Quantum-informed Tensor Adaptation

Since quantum circuits offer an elegant parameterization for large unitary matrices of shape $2^N \times 2^N$, by relaxing the unitarity constraint and allowing for arbitrary local dimensions, we can develop an effective tool for high-rank, parameter-efficient fine-tuning. Inspired by this, we propose **QuanTA**: **Quan**tum-informed **T**ensor **A**daptation, which parameterizes the parameter updates in a way analogous to a quantum circuit.

**Construction.** To illustrate the construction of QuanTA, we focus on the case of square weight matrices $W \in \mathbb{R}^{d \times d}$ in the main paper and defer the general case to Appendix B. In addition, we assume the hidden dimension $d$ can be decomposed as $d = d_1 \times d_2 \times \cdots \times d_N$ ‡. This condition is often satisfied for large language models. By reshaping $x \in \mathbb{R}^d$ to $x \in \mathbb{R}^{d_1 \times d_2 \times \cdots \times d_N}$, the hidden vector can be interpreted as a quantum state with $N$ "qudits," with the $n$th axis corresponding to a qudit with local dimension $d_n$.

Similar to a quantum circuit, QuanTA consists of "gates" (or tensors) that apply to only specific axes. Since single-axis gates are subsets of two-axis gates, it suffices to consider parameterizations using only two-axis gates. Let $T^{(\alpha)}$ be a tensor of shape $T^{(\alpha)} \in \mathbb{R}^{d_{m^{(\alpha)}} d_{n^{(\alpha)}} \times d_{m^{(\alpha)}} d_{n^{(\alpha)}}}$ that operates on the $m^{(\alpha)}$th and $n^{(\alpha)}$th axes with corresponding dimensions $d_{m^{(\alpha)}}$ and $d_{n^{(\alpha)}}$. Analogous to applying a two-qubit gate to a quantum state, applying this tensor to the hidden vector is defined as

$$(T^{(\alpha)}x)_{i_1,\ldots,i_m,\ldots,i_n,\ldots,i_N} := \sum_{j_m,j_n} T^{(\alpha)}_{i_m,i_n;j_m,j_n} x_{i_1,\ldots,j_m,\ldots,j_n,\ldots,i_N}, \tag{4}$$

where the $\alpha$ labels are dropped for simplicity, but it should be noted that different $T^{(\alpha)}$'s can be defined on different axes. Equivalently, this operation can be viewed as a matrix-vector multiplication with all but the $m^{(\alpha)}$th and $n^{(\alpha)}$th axes created as batch dimensions.

QuanTA is then constructed by sequentially applying a collection of such tensors $\{T^{(\alpha)}\}$ in the same manner as a quantum circuit:

$$\mathcal{T}x := \prod_\alpha T^{(\alpha)}x. \tag{5}$$

Although it is difficult to write the full Eq. (5) in index notation for an arbitrary set of tensors, we demonstrate in Appendix G that the `einsum` expression for this operation can be systematically generated.

As a concrete example of translating Eq. (5) to index notations and `einsum`, consider the case of $N = 3$; $\{T^{(\alpha)}\}$ consists of three tensors, each applied to two axes (as depicted in Fig. 1). In this case, it is easy to express in index notation the application of the QuanTA operator to the hidden vector;

$$(\mathcal{T}x)_{i_1,i_2,i_3} = \sum_{k_1,k_2} T^{(1)}_{i_1,i_2;k_1,k_2} \sum_{j_1,k_3} T^{(2)}_{k_1,i_3;j_1,k_3} \sum_{j_2,j_3} T^{(3)}_{k_2,k_3;j_2,j_3} x_{j_1,j_2,j_3} \tag{6}$$

as well as the calculation of the full QuanTA matrix;

$$\mathcal{T}_{i;j} = \mathcal{T}_{i_1,i_2,i_3;j_1,j_2,j_3} = \sum_{k_1,k_2} T^{(1)}_{i_1,i_2;k_1,k_2} \sum_{k_3} T^{(2)}_{k_1,i_3;j_1,k_3} T^{(3)}_{k_2,k_3;j_2,j_3}. \tag{7}$$

Although Eq. 6 and 7 may look complex in their formulation, in practice they can be easily implemented respectively using `einsum` as

---

‡Each $d_n$ does not need to be prime and the decomposition does not need to be unique

```
torch.einsum("...abc,efbc,diaf,ghde->...ghi", x, T_3, T_2, T_1)
```

and

```
torch.einsum("efbc,diaf,ghde->ghiabc", T_3, T_2, T_1)
```

**Initialization method.** At initialization, the adapted model should be the same as the base model and all the weight updates should be 0. However, enforcing $\mathcal{T}x = 0$ requires setting one or more $T^{(\alpha)} = 0$, impeding gradient propagation through the tensors and negatively impacting training performance.

To address this issue, we use another set of tensors $\{S^{(\alpha)}\}$ (with the corresponding QuanTA operator $\mathcal{S}$) that are initialized to the same value as $\{T^{(\alpha)}\}$ but remain frozen throughout fine-tuning. We then define the adapted layer as

$$y = W_\theta x := W_0 x + \mathcal{T}_\theta x - \mathcal{S}x, \tag{8}$$

where we use the subscript $\theta$ to denote tranable paraemters. At initialization, the terms $\mathcal{T}_\theta x$ and $-\mathcal{S}x$ exactly cancel out, ensuring the adapted layer reduces to the base model.

It is important to note that this initialization method does not introduce additional costs. After initialization, the full $\mathcal{S}$ matrix can be explicitly constructed, allowing us to redefine $W_0' = W_0 + \mathcal{S}$ and simplify the adapted layer to

$$y = W_\theta x = W_0' x + \mathcal{T}_\theta x. \tag{9}$$

## 6   Theoretical Results

Here, we list a few important theorem and provide the proofs in Appendix C

**Theorem 6.1** (Universality of QuanTA). *Let $W$ be an arbitrary matrix of shape $2^M \times 2^M$. For any collection of local dimensions $\{d_n\}$ such that each $d_n$ is a power of 2 and $\prod_n d_n = 2^M$, it is always possible to decompose $W$ into a finite sequence of tensors $\{T^{(\alpha)}\}$, where each tensor applies on two axes with local dimensions $d_{m^{(\alpha)}}$ and $d_{n^{(\alpha)}}$.*

We note that the fine-tuning method KronA [23] can be incorporated into our framework and considered as a special case of QuanTA.

**Theorem 6.2** (Rank representation). *Let $R = r(\mathcal{T})$ be the rank of the full QuanTA operator, $R^{(\alpha)} = r(T^{(\alpha)})$ be the rank of individual tensors, $d$ be the total dimension of $\mathcal{T}$, $d^{(\alpha)} = d_{m^{(\alpha)}} d_{n^{(\alpha)}}$ be the total dimension of the individual tensor $T^{(\alpha)}$, and $N_T$ be the total number of tensors. The following inequality always holds*

$$\sum_\alpha \frac{dR^{(\alpha)}}{d^{(\alpha)}} - d(N_T - 1) \le R \le \min_\alpha \frac{dR^{(\alpha)}}{d^{(\alpha)}}. \tag{10}$$

In the special case when all the tensors are full rank ($R^\alpha = d^{(\alpha)}$ for all $\alpha$), the full QuanTA operator is also full rank ($R = d$).

**Theorem 6.3** (Composition openness). *There exists a set $\mathbb{S} = \{\mathcal{M}_k\}$ of matrices generated from a fixed QuanTA structure and two matrices $\mathcal{M}_1, \mathcal{M}_2 \in \mathbb{S}$ such that $\mathcal{M}_1 \mathcal{M}_2 \notin \mathbb{S}$.*

We note that the composition openness condition is not satisfied by low-rank matrix decomposition because, for any two low-rank matrices of the same rank, their composition remains of the same rank. While LoRA may mitigate this limitation by introducing nonlinearity, its expressivity is still constrained by closure under composition. In contrast, QuanTA satisfies the composition openness condition even in the absence of nonlinearity, which suggests that its expressivity can continue to grow as the depth of the neural network increases, even if the network is nearly linear.

**No inference overhead**. As reparameterization-based methods, QuanTA does not impose any inference latency, since the trained $\mathcal{T}$ operator can be explicitly constructed as a matrix and merged into the base model weight matrix.

**Memory and computational complexity during fine-tuning.** In the forward pass, only a hidden vector of size $d$ is kept in the memory as we sequentially apply the tensors to it. Each tensor operation can be viewed as a batched matrix-vector multiplication and has a computational complexity of $d \cdot d_m d_n$ for tensor applying on the $m$th and $n$th axes, so the total computational complexity for a QuanTA layer is $d \cdot \sum_\alpha d_{m^{(\alpha)}} d_{n^{(\alpha)}}$. In addition, each tensor contains $(d_m d_n)^2$ elements. Therefore, each QuanTA layer contains $\sum_\alpha (d_{m^{(\alpha)}} d_{n^{(\alpha)}})^2$ trainable parameters that need to be stored in the optimizer. As an illustrative example, suppose $d_m = d^{1/N}$ for all $m$ and there is one tensor for every two axes, the computational complexity can be simplified to $N(N-1)/2 \cdot d^{1+2/N}$, and the parameter count becomes $N(N-1)/2 \cdot d^{4/N}$. When $N = 2$, QuanTA reduces to full fine-tuning.

# 7 Experiments

To benchmark QuanTA against other fine-tuning methods, we performed experiments on a wide range of datasets (see Appendix D for details). For all experiments, we avoid optimizing the hyperparameters on the test set. Instead, we create a validation set from the train set and optimize the hyperparameters on the validation set. All the results reported in this section are averaged over multiple experiments with varying random seeds, and the term "parameters" and "# params" in this section always refer to the trainable parameters. Details on the experiments and hyperparameters are shown in Appendix E.

| Model | PEFT Method | # Params (%) | $F_1$ Score ($\uparrow$) |
|---|---|---|---|
| | FT | 100% | 59.4 |
| | Series | 0.747% | 58.8 |
| | Parallel | 0.747% | 59.0 |
| LLaMA2$_{7B}$ | LoRA$_{r=8}$ | 0.062% | 54.0 |
| | LoRA$_{r=32}$ | 0.249% | 54.8 |
| | LoRA$_{r=128}$ | 0.996% | 56.2 |
| | QuanTA$_{16\text{-}8\text{-}8\text{-}4}$ **(Ours)** | 0.041% | 59.5 |
| | QuanTA$_{16\text{-}16\text{-}16}$ **(Ours)** | 0.261% | **59.6** |
| LLaMA2$_{13B}$ | LoRA$_{r=8}$ | 0.050% | 61.0 |
| | QuanTA$_{16\text{-}8\text{-}8\text{-}5}$ **(Ours)** | 0.029% | **69.0** |
| LLaMA2$_{70B}$ | LoRA$_{r=8}$ | 0.024% | 74.3 |
| | QuanTA$_{16\text{-}8\text{-}8\text{-}8}$ **(Ours)** | 0.014% | **79.4** |

Table 2: Benchmark of various fine-tuning methods on the DROP dataset using LLaMA2 7-70 billion parameter models as the base model. In each case, we report the average of $F_1$ score over 2-4 experiments with different random seeds.

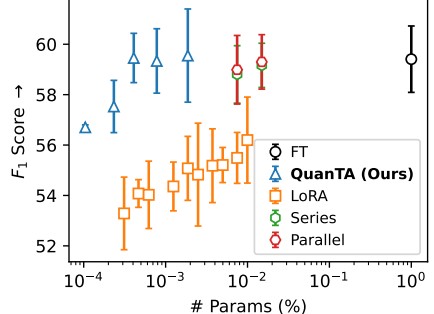

Figure 4: Benchmark of different fine-tuning methods on the DROP dataset as a function of training parameters using LLaMA2 7 billion parameter model as the base model.

**DROP Dataset**. We begin our benchmark with the DROP dataset [50], chosen as a representative example that requires high-rank fine-tuning. In Table 2, we compare our QuanTA method with LoRA of different ranks, as well as series and parallel adapters, by fine-tuning LLaMA2 [6] with up to 70 billion parameters.

As shown in Table 2, LoRA consistently underperforms compared to other fine-tuning methods. While increasing the rank improves performance, LoRA still falls short, suggesting the necessity of high-rank fine-tuning for this task. In addition, QuanTA achieves performance on par with, or better than, full fine-tuning using only a a small fraction of the parameters, demonstrating the effectiveness of QuanTA's high-rank fine-tuning capability.

To investigate how these methods scale with the number of trainable parameters, we conducted experiments varying the number of trainable parameters on LLaMA2-7B model. The results are

| Model | PEFT Method | # Params (%) | Accuracy (↑) | | | | | | | | |
|---|---|---|---|---|---|---|---|---|---|---|---|
| | | | **BoolQ** | **PIQA** | **SIQA** | **HellaS.** | **WinoG.** | **ARC-e** | **ARC-c** | **OBQA** | **Avg.** |
| GPT-3$_{175B}$* | – | – | 60.5 | 81.0 | – | 78.9 | 70.2 | 68.8 | 51.4 | 57.6 | – |
| PaLM$_{540B}$* | – | – | 88.0 | 82.3 | – | 83.4 | 81.1 | 76.6 | 53.0 | 53.4 | – |
| ChatGPT* | – | – | 73.1 | 85.4 | 68.5 | 78.5 | 66.1 | 89.8 | 79.9 | 74.8 | 77.0 |
| LLaMA$_{7B}$ | FT | 100% | 71.3 | 82.1 | 78.6 | 90.2 | 79.0 | 82.9 | 67.2 | 76.8 | 78.5 |
| | Prefix* | 0.11% | 64.3 | 76.8 | 73.9 | 42.1 | 72.1 | 72.9 | 54.0 | 60.6 | 64.6 |
| | Series* | 0.99% | 63.0 | 79.2 | 76.3 | 67.9 | 75.7 | 74.5 | 57.1 | 72.4 | 70.8 |
| | Parallel* | 3.54% | 67.9 | 76.4 | 78.8 | 69.8 | 78.9 | 73.7 | 57.3 | 75.2 | 72.3 |
| | LoRA* | 0.83% | 68.9 | 80.7 | 77.4 | 78.1 | 78.8 | 77.8 | 61.3 | 74.8 | 74.7 |
| | DoRA† | 0.43% | 70.0 | 82.6 | **79.7** | 83.2 | 80.6 | 80.6 | 65.4 | 77.6 | 77.5 |
| | DoRA† | 0.84% | 69.7 | **83.4** | 78.6 | 87.2 | 81.0 | 81.9 | 66.2 | 79.2 | 78.4 |
| | **QuanTA (Ours)** | 0.041% | **71.6** | 83.0 | **79.7** | **91.8** | **81.8** | **84.0** | **68.3** | **82.1** | **80.3** |
| LLaMA$_{13B}$ | Prefix* | 0.03% | 65.3 | 75.4 | 72.1 | 55.2 | 68.6 | 79.5 | 62.9 | 68.0 | 68.4 |
| | Series* | 0.80% | 71.8 | 83.0 | 79.2 | 88.1 | 82.4 | 82.5 | 67.3 | 81.8 | 79.5 |
| | Parallel* | 2.89% | 72.5 | 84.8 | 79.8 | 92.1 | 84.7 | 84.2 | 71.2 | 82.4 | 81.5 |
| | LoRA* | 0.67% | 72.1 | 83.5 | 80.5 | 90.5 | 83.7 | 82.8 | 68.3 | 82.4 | 80.5 |
| | DoRA† | 0.35% | 72.5 | 85.3 | 79.9 | 90.1 | 82.9 | 82.7 | 69.7 | 83.6 | 80.8 |
| | DoRA† | 0.68% | 72.4 | 84.9 | 81.5 | 92.4 | 84.2 | 84.2 | 69.6 | 82.8 | 81.5 |
| | **QuanTA (Ours)** | 0.029% | **73.2** | **85.4** | **82.1** | **93.4** | **85.1** | **87.8** | **73.3** | **84.4** | **83.1** |
| LLaMA2$_{7B}$ | FT | 100% | **72.9** | 83.0 | 79.8 | 92.4 | 83.0 | **86.6** | 72.0 | 80.1 | 81.2 |
| | LoRA† | 0.83% | 69.8 | 79.9 | 79.5 | 83.6 | 82.6 | 79.8 | 64.7 | 81.0 | 77.6 |
| | DoRA† | 0.43% | 72.0 | 83.1 | **79.9** | 89.1 | 83.0 | 84.5 | 71.0 | 81.2 | 80.5 |
| | DoRA† | 0.84% | 71.8 | 83.7 | 76.0 | 89.1 | 82.6 | 83.7 | 68.2 | 82.4 | 79.7 |
| | **QuanTA (Ours)** | 0.041% | 72.4 | **83.8** | 79.7 | **92.5** | **83.9** | 85.3 | **72.5** | **82.6** | **81.6** |
| LLaMA2$_{13B}$ | LoRA | 0.67% | 73.3 | 85.6 | 80.8 | 91.6 | 85.5 | 84.2 | 73.7 | 83.3 | 82.3 |
| | **QuanTA (Ours)** | 0.029% | **75.8** | **86.9** | **81.2** | **94.4** | **87.0** | **89.6** | **77.9** | **85.2** | **84.8** |
| LLaMA3$_{8B}$ | LoRA† | 0.70% | 70.8 | 85.2 | 79.9 | 91.7 | 84.3 | 84.2 | 71.2 | 79.0 | 80.8 |
| | DoRA† | 0.35% | 74.5 | 88.8 | 80.3 | **95.5** | 84.7 | 90.1 | 79.1 | **87.2** | 85.0 |
| | DoRA† | 0.71% | **74.6** | **89.3** | 79.9 | **95.5** | 85.6 | 90.5 | 80.4 | 85.8 | 85.2 |
| | **QuanTA (Ours)** | 0.035% | 74.3 | 88.1 | **81.8** | 95.1 | **87.3** | **91.1** | **81.7** | **87.2** | **85.8** |

Table 3: Benchmark on various commonsense reasoning tasks. All results of models and PEFT methods labeled with "*" are from [54], and results with "†" are from [20].

shown in Fig. 4. Each point in the figure represents an average of four experiments with different random seeds, and the standard deviation across these experiments is shown as error bars [§].

As illustrated in the figure, QuanTA achieves performance comparable to or better than full fine-tuning using a small fraction of trainable parameters. Conversely, LoRA only achieves subpar performance with a small number of trainable parameters, though its performance improves with an increase in parameters. Other PEFT methods, such as series and parallel adapters, achieve results close to full fine-tuning but use significantly more parameters than QuanTA.

**Commonsense Reasoning.** We continue to evaluate our method on a collection of commonsense reasoning datasets. Following the methodology in [54], we first fine-tune the model on the COM-MONSENSE170K dataset [54], a comprehensive collection of commonsense reasoning questions, and subsequently evaluate it on eight different downstream tasks.

In Table 3, we benchmark our QuanTA method against other fine-tuning techniques using 7- and 13-billion-parameter LLaMA and LLaMA2 models, as well as the 8-billion-parameter LLaMA3 model. Alongside prefix tuning, adapter methods, and LoRA, we also compare our approach to the recently proposed LoRA variant, the DoRA method [20]. The results clearly indicate that our QuanTA method outperforms LoRA in all cases and surpasses the DoRA method in most benchmarks, using less than one-tenth of the parameters.

**Arithmetic Reasoning**. We further test our method on arithmetic reasoning tasks by fine-tuning the model on MATH10K dataset [54] and assessing its performance on four tasks. We note that while [54] includes additional downstream tasks in the arithmetic reasoning benchmark, some test data was later found to have leaked into the training dataset. In this study, we only benchmark the four downstream

---

[§]We vary both the random seed for model initialization and the sampled train, dev, and test datasets, which could be the reason of large standard deviations.

| Model | PEFT Method | # Params (%) | Accuracy ($\uparrow$) | | | | |
|---|---|---|---|---|---|---|---|
| | | | AQuA | GSM8K | MAWPS | SVAMP | Avg. W/O AQuA |
| GPT-3.5$_{175B}$* | – | – | 38.9 | 56.4 | 87.4 | 69.6 | 71.1 |
| LLaMA2$_{7B}$ | FT | 100% | *19.3* | 65.2 | 92.0 | 80.7 | 79.3 |
| | LoRA | 0.83% | *17.5* | 65.7 | 91.2 | **80.8** | 79.6 |
| | **QuanTA (Ours)** | 0.19% | *16.7* | **67.0** | **94.3** | 80.3 | **80.5** |
| LLaMA2$_{13B}$ | LoRA | 0.67% | *16.7* | 72.3 | 90.8 | 84.3 | 82.5 |
| | **QuanTA (Ours)** | 0.13% | *18.9* | **72.4** | **94.5** | **84.8** | **83.9** |

Table 4: Benchmark on various arithmetic reasoning tasks. GPT-3.5 (labeled with "*") results are taken from [54].

tasks unaffected by this data leakage. Additionally, our evaluation procedure differs slightly from that in [54] (see Appendix E for details).

Table 4 presents the evaluation results on the four downstream tasks. Notably, all questions in the AQuA dataset are multiple-choice with mostly five options, and all models except GPT-3.5 failed to achieve accuracy higher than 20%. Therefore, we conclude that all models perform equally poorly on this task and exclude it from the average accuracy computation. This phenomenon is also consistent with previous findings [54, 20]. The results show that QuanTA significantly outperforms LoRA and even surpasses full fine-tuning with a small number of parameters. It is surprising that QuanTA exceeds full fine-tuning in these tasks, which may be due to overfitting or the challenges of optimizing hyperparameters for full fine-tuning.

In Appendix F, we include benchmarks with additional fine-tuning methods and on additional datasets.

**Limitations.** QuanTA currently requires applying the tensors sequentially to the hidden vectors, which may result in underutilizing the GPU when the tensors are too small. It will be helpful to develop a more efficient implementation to fully utilize GPU resources. The hyperparameters in QuanTA, such as the number of tensors applying on the same axes, have not been optimized. Choosing an optimal set of tensors could further enhance the performance of QuanTA. In the current experiments, we only consider LLaMA model series and a thorough study on different models will be beneficial if more computational resources are available.

# 8    Conclusion

In this paper, we introduced QuanTA, a novel, easy-to-implement, PEFT method with no inference overhead for large language models. QuanTA leverages quantum-inspired techniques to achieve high-rank adaptations, addressing the limitations of existing low-rank methods. QuanTA introduces high-rank fine-tuning through the universality theorem and rank representation theorem. Our extensive experiments demonstrate the efficacy of QuanTA across various tasks, including commonsense reasoning, arithmetic reasoning, and scalability. QuanTA consistently outperforms traditional fine-tuning methods and other PEFT approaches, achieving superior performance with a significantly smaller number of trainable parameters. This highlights the potential of quantum-informed techniques in enhancing the adaptability and efficiency of large language models.

QuanTA offers a scalable and efficient solution for fine-tuning large language models, advancing the state-of-the-art in natural language processing. There are several promising directions for future research and development of QuanTA. Expanding its application to a wider range of tasks and specialized domains could demonstrate its versatility and robustness. Combining QuanTA with other PEFT methods or incorporating it into ensemble models might further enhance performance, particularly for complex tasks. The parameter efficiency of QuanTA may also imply a lower chance of overfitting. Additionally, exploring advanced optimization techniques tailored specifically for QuanTA could improve convergence rates and overall efficiency. Further design based on principles from quantum computing, such as entanglement and superposition, may lead to even more efficient fine-tuning methods. Our work paves the way for further exploration of quantum-informed methods or even future quantum technologies for machine learning, making it a valuable approach for both research and practical applications with broader impacts.

## Broader Impacts

The development of QuanTA represents an important advancement in the fine-tuning of LLMs, with profound societal implications. By leveraging quantum-informed methods, QuanTA reduces computational and memory demands, making advanced NLP capabilities more accessible and cost-effective. This democratization of AI technology can facilitate its adoption in resource-constrained environments, bridging technological disparities. Additionally, the integration of quantum techniques could spark interdisciplinary innovations, enhancing healthcare diagnostics, financial risk assessment, and personalized education. Furthermore, QuanTA's efficiency aligns with global sustainability efforts by reducing the energy consumption associated with AI training, contributing to the reduction of AI's carbon footprint. Thus, QuanTA not only advances NLP but also promotes inclusive, sustainable, and impactful AI technologies across various sectors. However, the deployment of such powerful AI models raises concerns about data privacy, security, and the potential misuse of AI technologies. Addressing these ethical and societal challenges is crucial to ensure that the benefits of QuanTA are realized responsibly and equitably.

## Acknowledgements

The authors acknowledge support from the National Science Foundation under Cooperative Agreement PHY-2019786 (The NSF AI Institute for Artificial Intelligence and Fundamental Interactions, http://iaifi.org/). This material is based upon work supported by the U.S. Department of Energy, Office of Science, National Quantum Information Science Research Centers, Co-design Center for Quantum Advantage (C2QA) under contract number DE-SC0012704. The research was sponsored by the United States Air Force Research Laboratory and the Department of the Air Force Artificial Intelligence Accelerator and was accomplished under Cooperative Agreement Number FA8750-19-2-1000. The computations in this paper were run on the FASRC cluster supported by the FAS Division of Science Research Computing Group at Harvard University.

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

# Appendix

## A  Additional Details on Subspace Similarity

In the main paper, we use the subspace similarity to measure the "intrinsic rank" of fine-tuning on a specific dataset. In this section, we provide more details on it.

While it is tempting to compute the rank by performing singular value decomposition (SVD) on the weight matrices of fully fine-tuned models, such measurement generally overestimates the intrinsic rank due to random parameter updates during fine-tuning that are irrelevant to the performance on the downstream tasks. The authors of [10] proposes a better way to measure the intrinsic rank, which we describe as follows.

First, we run LoRA fine-tuning for two different ranks $r_1$ and $r_2$ and obtain the LoRA weight updates $\Delta W^{(r_1)} = B^{(r_1)} A^{(r_1)}$ and $\Delta W^{(r_2)} = B^{(r_2)} A^{(r_2)}$. Then, we perform singular value decompositions on the weights to obtain $\Delta W^{(r)} = U^{(r)} S^{(r)} V^{(r)\top}$. Next, let's denote the first $i$ right singular vectors of $\Delta W^{(r_1)}$ (first $i$ columns of $V^{(r_1)}$ as $V^{(r_1,i)}$) and the first $j$ right singular vectors of $\Delta W^{(r_2)}$ (first $j$ columns of $V^{(r_2)}$) as denoted as $V^{(r_2,j)}$. The subspace similarity between these two subspace is defined as

$$\phi(r_1, r_2, i, j) = \frac{\left\| V^{(r_1,i)\top} V^{(r_2,j)} \right\|_F^2}{\min(i,j)} \in [0,1]. \tag{A.1}$$

This function equals 1 if any subspace can be contained in the other, equals 0 if the two subspaces are orthogonal, and in general measures the overlap between 0 and 1 between the two subspaces. FOr fine-tuning that has a low "intrinsic rank"; only the subspace spanned by the first few singular vectors (that correspond to the largest few singular vectors) should be similar, with the rest nearly perpendicular originating from random noise during fine-tuning. Thus, the subspace similarity should be close to 1 only when either $i$ or $j$ is small, and quickly decays to 0 for large $i$ and $j$. On the other hand, when the "intrinsic rank" is high, all the singular vectors in one subspace can be important and therefore would appear in the other. In this case, the subspace similarity can remain high for all values of $i$ and $j$.

In the main paper, we choose $r_1 = 64$ and $r_2 = 128$, and measure the subspace similarity for both the RTE dataset [49] and the DROP dataset [50], and reported the values corresponding to the query weight matrix of the 16th layer. In this section, we include results corresponding to additional weight matrices. In Fig. A.1 and A.2, we show the subspace similarities for the value weight matrix at layer 16 and 23. We observe that the similar behaviors appear for these two weight matrices as in the main paper, where the RTE dataset exhibits a low "intrinsic-rank", while the DROP dataset has a high "intrinsic-rank".

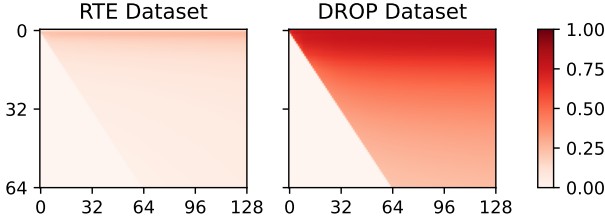

Figure A.1: Subspace similarities between two LoRA experiments of different ranks (64 and 128) for two datasets for the value weight matrix at layer 16.

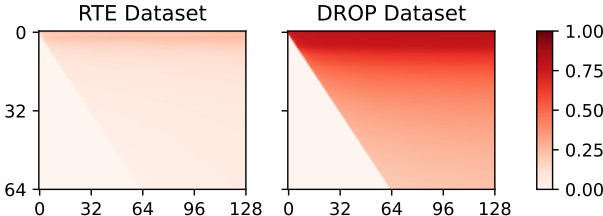

Figure A.2: Subspace similarities between two LoRA experiments of different ranks (64 and 128) for two datasets for the value weight matrix at layer 23.

# B  Constructing General QuanTA Operators

In the main paper, we discussed QuanTA operation when the layer weight is square $W_0 \in \mathbb{R}^{d \times d}$ and we assumed $d$ to be a composite number that can be decomposed into $d = d_1 \times d_2 \times \cdots \times d_N$. Here, we will consider more general cases.

Let's first consider general rectangular matrices $W_0 \in \mathbb{R}^{d \times k}$ but still assume $d$ and $k$ to permit decompositions in the form $d = d_1 \times d_2 \times \cdots \times d_N$ and $k = k_1 \times k_2 \times \cdots \times k_N$, without loss of generality, let's assume $d \geq k$ (transpose the matrix if $d < k$). In addition, assume $d$ and $k$ has a simple ratio and let $d_1/k_1 = d/k$. Note that this requirement may seem strict, but in many cases, $d$ can be as simple as a multiple of $k$. (For example, LLaMA2-70B model contains many $1024 \times 8192$ and $8192 \times 1024$ weight matrices, in which case $d/k = 8/1$). Then, we can set $d_n = k_n$ for all $n > 1$. Among the collection of tensors, let's assume $T^\alpha \in \mathbb{R}^{d_1 d_n \times k_1 d_n}$ ($d_n = k_n$) is a "rectangular" tensor that applies on the first and $n$th axes. After applying this tensor, the hidden vector changes shape from $\mathbb{R}^{d_1 \times d_2 \times \cdots \times d_N}$ to $\mathbb{R}^{k_1 \times d_2 \times \cdots \times d_N}$, making the hidden vector into the correct size. Then, one just needs to make sure that all the tensors subsequent to this tensor needs to have the correct shape when applying to the first axis. A pictorial representation is shown in Fig. B.3.

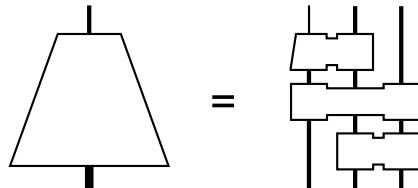

Figure B.3: Illustration of how to parameterize rectangular matrix with QuanTA

In the special case where the "rectangular" tensor is the last tensor applied to the hidden vector, this operation can be equivalently written as applying a "square" tensor of shape $\mathbb{R}^{d_1 d_n \times d_1 d_n}$, resulting in an output hidden vector of shape $y \in \mathbb{R}^{d_1 \times d_2 \times \cdots \times d_N} \simeq \mathbb{R}^d$ and then slicing the first $k$ elements from it. In the case when $d < k$, this operation needs to be reversed, and it can be expressed as first padding the hidden vector to appropriate size, before applying the tensor circuit.

More generally, it is possible to choose any set of $\{d_n\}_{n=1}^N$ and $\{k_n\}_{n=1}^N$ (where there products may or may not equal to $d$ and $k$), as well as any of tensors that transforms $\mathbb{R}^{d_1 \times d_2 \times \cdots \times d_N} \to \mathbb{R}^{k_1 \times k_2 \times \cdots \times k_N}$. Then, one can always truncate or pad the input vector of size $d$ to length $\prod_{n=1}^N d_n$, and the output vector from size $\prod_{n=1}^N k_n$ to $k$. We note that while this method will always work, it is recommended to choose $\{d_n\}_{n=1}^N$ and $\{k_n\}_{n=1}^N$ such that there products are close to $d$ and $k$, to achieve the best performance and avoid unnecessary cost. We furhter note that having $d \neq \prod_{n=1}^N d_n$ and $k \neq \prod_{n=1}^N k_n$ still allows the full QuanTA operator $\mathcal{T}$ to be merged into the original weight matrix, by padding and truncating the $\mathcal{T}$ into $\mathbb{R}^{d \times k}$.

# C Additional Theoretical Results and Proofs

In the main paper, we have listed a few important theorems. In this section, we provide the proof for these theorems. We will first need the following lemma, which is fundamental to modern quantum computation. The proof of this lemma can be found in any modern quantum computation textbook such as Ref. [53] or in Ref. [55, 56].

**Lemma C.1** (Universality of two-qubit gates). *Any $2^M \times 2^M$ unitary matrix can be written as a quantum circuit using just two-qubit gates of size $4 \times 4$.*

This immediately gives us the following corollary.

**Corollary C.1.** *Any $\prod_n d_n \times \prod_n d_n$ unitary matrix with $d_n$ a power of 2 can be written as a quantum circuit using two-qudit gates of size $d_m d_n \times d_m d_n$.*

*Proof.* It is possible to reshape the matrix into $2^M \times 2^M$ and use Lemma C.1 to obtain a two-qubit gates representation. Since two-qubit gates are a subset of two-qudit gates, this already concludes the proof. However, one can group the two-qubit gates that apply to the same qudit to reduce reduce the gate count. $\square$

We also need another lemma from quantum computation [53].

**Lemma C.2** (Phase Rotation). *For any diagonal unitary matrix of size $2^M \times 2^M$ (whose elements are in the form of $e^{i\theta_k}$), there exists a finite sequence of two-qubit gates with parameters $\{\theta_k\}$, where the structure of the sequence is fixed for all possible set of $\{\theta_k\}$ and each two-qubit gate is an analytic function of $\{\theta_k\}$, that can exactly represent the diagonal matrix.*

Similar to Corollary C.1, Lemma C.2 can also be extended to

**Corollary C.2.** *For any diagonal unitary matrix of size $\prod_n d_n \times \prod_n d_n$ with $d_n$ a power of 2, there always exists a finite sequence of two-qudit gates with parameters $\{\theta_k\}$, where the structure of the sequence is fixed for all possible set of $\{\theta_k\}$ and each two-qudit gate is an analytic function of $\{\theta_k\}$, that can exactly represent the diagonal matrix.*

We can analytically continue Corollary C.2 to nonunitary diagonal matrices.

**Corollary C.3.** *For any diagonal matrix of size $\prod_n d_n \times \prod_n d_n$ with $d_n$ a power of 2 with diagonal elements $\{a_k\}$, there exists a finite sequence of two-qudit gates with parameters $\{a_k\}$, where the structure of the sequence is fixed for all possible set of $\{\theta_k\}$, that can exactly represent the diagonal matrix.*

*Proof.* Consider Corollary C.2, since both the full unitary matrix and the sequence of two-qudit gates are finite, and are analytic functions of $\{\theta_k\}$, their analytic continuation must also be equal. Therefore, setting $\theta_k$'s to be imaginary numbers concludes the proof. $\square$

Now, we are finally ready to prove the universality theorem.

**Theorem C.1** (Universality of QuanTA). *Let $W$ be an arbitrary matrix of shape $2^M \times 2^M$. For any collection of local dimensions $\{d_n\}$ such that each $d_n$ is a power of 2 and $\prod_n d_n = 2^M$, it is always possible to decompose $W$ into a finite sequence of tensors $\{T^{(\alpha)}\}$, where each tensor applies on two axes with local dimensions $d_{m^{(\alpha)}}$ and $d_{n^{(\alpha)}}$.*

*Proof.* Let $U$, $S$ and $V$ be the singular value decomposition of $W$. Since $U$ and $V$ are unitary matrices, it immediately follows from Corollary C.1 that they can be written as a finite sequence of tensors. In addition, since $S$ is a diagonal matrix, Corollary C.3 shows that it can also be written as a finite sequence of tensors. Combining all tensors into the same QuanTA operator by applying the sequentially, we obtain the full QuanTA decomposition of $W$. $\square$

**Theorem C.2** (Rank representation). *Let $R = r(\mathcal{T})$ be the rank of the full QuanTA operator, $R^{(\alpha)} = r(T^{(\alpha)})$ be the rank of individual tensors, $d$ be the total dimension of $\mathcal{T}$, $d^{(\alpha)} = d_{m^{(\alpha)}} d_{n^{(\alpha)}}$ be the total dimension of the individual tensor $T^{(\alpha)}$, and $N_T$ be the total number of tensors. The following inequality always holds*

$$\sum_\alpha \frac{dR^{(\alpha)}}{d^{(\alpha)}} - d(N_T - 1) \le R \le \min_\alpha \frac{dR^{(\alpha)}}{d^{(\alpha)}}. \tag{C.2}$$

*Proof.* The rank of the product of two matrices $A$ and $B$ of shape $d \times d$ satisfies the inequality $r(A) + r(B) - d \leq r(AB) \leq \min\{r(A), r(B)\}$. In QuanTA, each tensor can be viewed as a large matrix, where $T^{(\alpha)}$ is applied on the $m^{(\alpha)}$th and $n^{(\alpha)}$th axes, and identity matrix is applied to the rest of the axes. In this case, the rank of this operation is the same as the rank of $T^{(\alpha)}$ times the rank of the product of the identity matrices, which equals $\frac{dR^{(\alpha)}}{d^{(\alpha)}}$. Then, using the above inequality multiple times concludes our proof. $\square$

**Theorem C.3** (Composition openness). *There exists a set* $\mathbb{S} = \{\mathcal{M}_k\}$ *of matrices generated from a fixed QuanTA structure and two matrices* $\mathcal{M}_1, \mathcal{M}_2 \in \mathbb{S}$ *such that* $\mathcal{M}_1 \mathcal{M}_2 \notin \mathbb{S}$.

*Proof.* We consider a set of matrices $\mathbb{S}$ generated by the QuanTA structure that consists of one layer of single-qubit rotation gates followed by a layer of two-qubit CNOT gates [53] and then one layer of single-qubit rotation gates.[¶] This is a set of unitary matrices with entanglement generation determined by the number of layers of CNOT gates. Consider $\mathcal{M}_1, \mathcal{M}_2 \in \mathbb{S}$, according to quantum information theory, it is not possible to have $\mathcal{M}_1 \mathcal{M}_2 \in \mathbb{S}$. This is because $\mathcal{M}_1 \mathcal{M}_2$ has two layers of CNOT gates which can generate more entanglement than any element $\mathcal{M}_3 \in \mathbb{S}$ that only contains one layer of CNOT gates. $\square$

# D   Datasets

| Dataset | Task | # Train | # Val | # Test | Metric | Answer |
|---|---|---|---|---|---|---|
| DROP [50] | Reading comprehension with discrete reasoning | 2000 | 800 | 1200 | $F_1$-Score | Phrase |
| COMMONSENSE170K [54] | Commonsense reasoning (Train) | 170020 | 400 | – | – | – |
| BoolQ [57] | Commonsense reasoning (Test) | – | – | 3270 | Accuracy | Yes/No |
| PIQA [58] | Commonsense reasoning (Test) | – | – | 1838 | Accuracy | Option |
| SIQA [59] | Commonsense reasoning (Test) | – | – | 508 | Accuracy | Option |
| HellaSwag [60] | Commonsense reasoning (Test) | – | – | 10042 | Accuracy | Option |
| WinoGrande [61] | Commonsense reasoning (Test) | – | – | 1267 | Accuracy | Option |
| ARC-Easy [62] | Commonsense reasoning (Test) | – | – | 2376 | Accuracy | Option |
| ARC-Challenge [62] | Commonsense reasoning (Test) | – | – | 1172 | Accuracy | Option |
| OBQA [63] | Commonsense reasoning (Test) | – | – | 500 | Accuracy | Option |
| MATH10K [54] | Arithmetic reasoning (Train) | 9519 | 400 | – | – | – |
| AQuA [64] | Arithmetic reasoning (Test) | – | – | 254 | Accuracy | Option |
| GSM8K [65] | Arithmetic reasoning (Test) | – | – | 1319 | Accuracy | Number |
| MAWPS [66] | Arithmetic reasoning (Test) | – | – | 238 | Accuracy | Number |
| SVAMP [67] | Arithmetic reasoning (Test) | – | – | 1000 | Accuracy | Number |

Table D.1: List of datasets used in this work.

In this section, we describe the datasets used in this paper. In Table D.1, the list of datasets used in this paper is listed.

For the DROP dataset [50], we subsample 2000 samples from the original train set as our train set, 800 samples from the train set as our validation set, and 1200 samples from the original validation set as our test set, since the original dataset does not contain a test set on Hugging Face. In addition, the $F_1$-score is used to measure the closeness of the models' output compared to the ground truth since it is in general a phrase.

For all commonsense reasoning tasks, we first fine-tune a single model on the COMMONSENSE170K dataset collected by [54], and evaluate the same model on eight different commonsense reasoning tasks [57–63], which we use the version provided by [54]. The COMMONSENSE170K dataset is split into a train set of 170020 samples, and a validation set of 400 samples. All of the commonsense reasoning tasks are either Yes/No questions or multiple choice questions. In these tasks, the model is asked to choose the best answer from all the options, and accuracy is used as the evaluation metric.

For all arithmetic reasoning tasks, we fine-tune a single model on the MATH10K dataset [54] and evaluate the same model on four different tasks [64–67]. We split the MATH10K dataset into a train

---

[¶]Note that the single-qubit gates can be absorbed into the two-qubit gates, so this structure is within our QuanTA framework.

set of 9519 samples, and a validation set of 400 samples. Similar to the commonsense reasoning tasks, we use the version of the datasets provided by [54]. In addition, in [54], there was found some data leak issues in some of the arithmetic datasets. Here, we only consider the datasets that are unaffected. In the arithmetic reasoning tasks, although the model is asked to generate the step-by-step solution for the final answer, only the final answer is parsed to measure the accuracy. For AQuA, we parse the output text to find the last character such that it is one of the options. For the other three tasks which require numerical answers, we simply parse the last number from the output text, and consider the answer to be correct if it is the same as the ground truth for up to 4 decimal places.

# E    Hyperparameters and Experimental Details

In this section, we describe the hyperparameter choices and the experimental details. All the experiments are conducted on NVIDIA A100 GPUs with 80 GB memory. GPU count used in each experiment will be explained later. The code used to produce the experiments is released on GitHub at `https://github.com/quanta-fine-tuning/quanta`. Our code is implemented using [54] and [68] as references.

## E.1    QuanTA parameterization

Although QuanTA supports decomposing the hidden dimension into an arbitrary number of axes $N$ and a wide selection of collections of tensors $\{T^{(\alpha)}\}$ as long as the tensors are compatible with the axes, in this work, we focus on $N = 3$, $4$ and $5$, where exactly one tensor is applied on each unique combination of axes. For example, there are 3 tensors when $N = 3$, 6 tensors when $N = 4$, and 10 tensors when $N = 5$. Note that if $N = 2$, there is only a single tensor and this approach reduces to the full fine-tuning. Since these tensors are applied sequentially, and matrix multiplications in general don't commute, the order of tensor application can also affect the result. In the case of $N = 3$, the QuanTA layer is constructed as Fig. 1 in the main paper. For $N = 4$ and $N = 5$, we show the construction in Fig. E.4

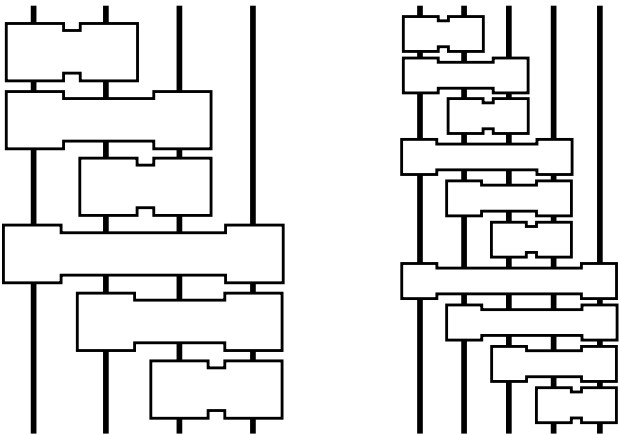

Figure E.4: QuanTA architecture used in this work for $N = 4$ and $N = 5$.

## E.2    Experiments on DROP dataset

In Table E.2, we show the hyperparameters used for the DROP experiments. Only LoRA and QuanTA are applied to the 13- and 70-billion-parameter LLaMA2 models. For the 13-billion-parameter model or smaller, only a single A100 GPU is used for fine-tuning. And for the 70-billion-parameter model, four A100 GPUs are used. For all experiments, the hyperparameters are only optimized on the 7-billion-parameter LLaMA2 model, and applied directly on larger models. In addition, all the hyperparameters are optimized on the validation set, before evaluating the model on the test set. We further note that we choose the best checkpoint obtained during fine-tuning, in terms of the $F_1$-score on the validation set, as the final model to apply on the test set. Because of this, the number-of-epoch parameter does not introduce a significant effect to the final result, as long as the training converges.

Therefore, this hyperparameter is chosen rather arbitrarily between 3 and 6. We further note that the batch sizes reported here are the effective batch sizes, including the gradient accumulation steps.

### E.3  Experiments on commonsense datasets

We explain the details of the experiments on commonsense datasets. As mentioned before, in this experiment, we first fine-tune the model on the joint COMMONSENSE170K dataset and evaluate the same fine-tuned model on all downstream tasks. Similar to the drop dataset, we optimize the hyperparameters on the validation set that we created from the COMMONSENSE170K dataset and choose the best checkpoint in terms of the validation accuracy to evaluate on the benchmarks. The hyperparameters are listed in Table E.3.

### E.4  Experiments on arithmetic datasets

We further explain the details of the experiments on arithmetic datasets. Similar to previous, we first fine-tune the model on the joint MATH10K dataset and evaluate the same fine-tuned model on all downstream tasks and we optimize the hyperparameters on the validation set that we created from the MATH10K dataset and choose the best checkpoint in terms of the validation accuracy to evaluate on the benchmarks. The hyperparameters are listed in Table E.4. Notice that we choose a different set of module for LoRA to match the experimental setup of [54, 20].

| Experiment | Hyperparameters | Values |
|---|---|---|
| **FT** | Batch Size | {2, 4, 8} |
| | Optimizer | AdamW |
| | Scheduler | Linear Scheduler |
| | Learning Rate | {5e-6, 1e-5} |
| | Weight Decay | 0 |
| | Dropout | 0 |
| | Number of GPUs | 1 |
| **Series Adapters** | Batch Size | 4 |
| | Optimizer | AdamW |
| | Scheduler | Linear Scheduler |
| | Learning Rate | 1e-4 |
| | Weight Decay | 0 |
| | Dropout | 0 |
| | Bottleneck | [64, 128] |
| | Modules | Default |
| | Number of GPUs | 1 |
| **Parallel Adapters** | Batch Size | 4 |
| | Optimizer | AdamW |
| | Scheduler | Linear Scheduler |
| | Learning Rate | 1e-4 |
| | Weight Decay | 0 |
| | Dropout | 0 |
| | Bottleneck | [64 ,128] |
| | Modules | Default |
| | Number of GPUs | 1 |
| **LoRA** | Batch Size | 4 |
| | Optimizer | AdamW |
| | Scheduler | Linear Scheduler |
| | Learning Rate | {1e-4, 2e-4, 5e-4, 6e-4} |
| | Weight Decay | 0 |
| | Dropout | 0 |
| | $r$ | [4, 6, 8, 16, 24, 32, 48, 64, 96, 128] |
| | $\alpha$ | 16 |
| | Modules | (q_proj v_proj) (Same as Default) |
| | Number of GPUs | [1, 4] |
| **QuanTA** | Batch Size | 4 |
| | Optimizer | AdamW |
| | Scheduler | Linear Scheduler |
| | Learning Rate | 1e-4 |
| | Weight Decay | 0 |
| | Dropout | 0 |
| | $N$ | [3, 4, 5] |
| | $d_1$-$d_2$-$\cdots$-$d_N$ | [8-8-4-4, 8-8-8-8, 16-8-8-4, 16-16-4-4, 16-16-16, 16-8-8-5, 16-8-8-8] |
| | Modules | (q_proj v_proj) |
| | Number of GPUs | [1, 4] |

Table E.2: Hyperparameters used for DROP dataset for various fine-tuning methods. Curly brackets include the hyperparameter values tested during hyperparameter optimization, with the actual hyperparameter(s) underscored. Square brackets include hyperparameter values for different experiments conducted in the main paper.

| Experiment | Hyperparameters | Values |
|:---:|:---:|:---:|
| **FT** | Batch Size | 4 |
| | Optimizer | AdamW |
| | Scheduler | Linear Scheduler |
| | Learning Rate | 1e-5 |
| | Weight Decay | 0 |
| | Dropout | 0 |
| | Number of GPUs | 1 |
| **QuanTA** | Batch Size | 4 |
| | Optimizer | AdamW |
| | Scheduler | Linear Scheduler |
| | Learning Rate | {5e-5, 1e-4} |
| | Weight Decay | 0 |
| | Dropout | 0 |
| | $N$ | 4 |
| | $d_1$-$d_2$-$\cdots$-$d_N$ | [16-8-8-4, 16-8-8-5] |
| | Modules | (q_proj v_proj) |
| | Number of GPUs | 1 |

Table E.3: Hyperparameters used for commonsense experiments. Curly brackets include the hyperparameter values tested during hyperparameter optimization, with the actual hyperparameter(s) underscored. Square brackets include hyperparameter values for different experiments conducted in the main paper.

| Experiment | Hyperparameters | Values |
|:---:|:---:|:---:|
| **FT** | Batch Size | 4 |
| | Optimizer | AdamW |
| | Scheduler | Linear Scheduler |
| | Learning Rate | 1e-5 |
| | Weight Decay | 0 |
| | Dropout | 0 |
| | Number of GPUs | 1 |
| **LoRA** | Batch Size | 4 |
| | Optimizer | AdamW |
| | Scheduler | Linear Scheduler |
| | Learning Rate | {1e-4, 3e-4} |
| | Weight Decay | 0 |
| | Dropout | 0 |
| | $r$ | 32 |
| | $\alpha$ | 16 |
| | Modules | (q_proj k_proj v_proj up_proj down_proj) |
| | Number of GPUs | 1 |
| **QuanTA** | Batch Size | 4 |
| | Optimizer | AdamW |
| | Scheduler | Linear Scheduler |
| | Learning Rate | {1e-4, 3e-4} |
| | Weight Decay | 0 |
| | Dropout | 0 |
| | $N$ | 4 |
| | $d_1\text{-}d_2\text{-}\cdots\text{-}d_N$ | {16-8-8-4, 16-16-4-4, 16-8-8-5, 16-16-5-4} |
| | Modules | {(q_proj v_proj), 
 (q_proj k_proj v_proj up_proj down_proj)} |
| | Number of GPUs | 1 |

Table E.4: Hyperparameters used for arithmetic experiments. Curly brackets include the hyperparameter values tested during hyperparameter optimization, with the actual hyperparameter(s) underscored. Square brackets include hyperparameter values for different experiments conducted in the main paper.

# F  Additional Benchmarking Results

In this section, we include benchmarking results with additional fine-tuning methods and on additional datasets. In Table F.5 and F.6, we include additional comparisons to MoRA [31], LoRETTA [13], and KronA [23]. In Table F.7, we include additional results on five commonsense understanding tasks from the GLUE benchmark [69] using RoBERTa model [70].

| PEFT Method | # Params (%) | $F_1$ Score (↑) |
|---|---|---|
| FT | 100% | 59.4 |
| Series | 0.747% | 58.8 |
| Parallel | 0.747% | 59.0 |
| LoRA$_{r=8}$ | 0.062% | 54.0 |
| LoRA$_{r=32}$ | 0.249% | 54.8 |
| LoRA$_{r=128}$ | 0.996% | 56.2 |
| *MoRA$_{r=8}$* | 0.062% | 58.6 |
| *MoRA$_{r=32}$* | 0.249% | 58.2 |
| *MoRA$_{r=128}$* | 0.996% | 58.9 |
| *LoRETTA$_{r=8}$* | 0.009% | 48.6 |
| *LoRETTA$_{r=32}$* | 0.083% | 54.9 |
| *LoRETTA$_{r=128}$* | 1.254% | 59.1 |
| *KronA$_{64\text{-}64}$* | 0.008% | 50.9 |
| *KronA$_{256\text{-}16}$* | 0.062% | 57.7 |
| *KronA$_{1024\text{-}4}$* | 0.996% | 58.5 |
| **QuanTA$_{16\text{-}8\text{-}8\text{-}4}$ (Ours)** | 0.041% | 59.5 |
| **QuanTA$_{16\text{-}16\text{-}16}$ (Ours)** | 0.261% | **59.6** |

Table F.5: Benchmark of various fine-tuning methods on the DROP dataset using LLaMA2 7 billion parameter model as the base model. Fine-tuning methods in addition to the main paper are shown in italic font. In each case, we report the average of $F_1$ score over 4 experiments with different random seeds. For LoRA, MoRA and LoRETTA, the subscript labels the rank; for KronA the subscript labels the sizes of the matrices; and for QuanTA, the subscript labels the dimensions of axes.

| PEFT Method | # Params (%) | Accuracy (↑) | | | | | | | | |
|---|---|---|---|---|---|---|---|---|---|---|
| | | BoolQ | PIQA | SIQA | HellaS. | WinoG. | ARC-e | ARC-c | OBQA | Avg. |
| LoRA[†] | 0.70% | 70.8 | 85.2 | 79.9 | 91.7 | 84.3 | 84.2 | 71.2 | 79.0 | 80.8 |
| DoRA[†] | 0.35% | 74.5 | 88.8 | 80.3 | **95.5** | 84.7 | 90.1 | 79.1 | **87.2** | 85.0 |
| DoRA[†] | 0.71% | **74.6** | **89.3** | 79.9 | **95.5** | 85.6 | 90.5 | 80.4 | 85.8 | 85.2 |
| *LoRETTA* | 0.13% | 74.3 | 87.5 | 80.9 | 94.5 | 86.7 | **92.1** | 81.5 | 85.8 | 85.4 |
| *KronA* | 0.052% | 72.9 | 87.1 | 80.6 | 92.1 | 85.1 | 87.8 | 76.0 | 84.3 | 83.2 |
| **QuanTA (Ours)** | 0.035% | 74.3 | 88.1 | **81.8** | 95.1 | **87.3** | 91.1 | **81.7** | **87.2** | **85.8** |

Table F.6: Benchmark on various commonsense reasoning tasks using LLaMA3 8 billion parameter model as the base model. Fine-tuning methods in additional to the main paper are shown in italic font. All results of models and PEFT methods labeled with "*" are from [54], and results with "†" are from [20].

| PEFT Method | # Params (%) | Accuracy (↑) | | | | |
|---|---|---|---|---|---|---|
| | | SST-2 | MRPC | CoLA | RTE | STS-B |
| LoRA | 0.71% | **94.01** | 91.48 | **62.08** | 74.51 | 90.48 |
| **QuanTA (Ours)** | 0.62% | 93.81 | **91.67** | **62.08** | **77.26** | **90.68** |

Table F.7: Benchmark on five natural language understanding tasks using RoBERTa model as the base model.

# G  Systematical Way to Generate `einsum` Expressions

In the main paper, we show an example of how to implement QuanTA operation easily using `einsum`. Here, we show how to systematically generate the `einsum` expression more generally. For illustrative purposes, we focus on the case where there is exactly one tensor for every combination of two axes.

First, we show how to generate the `einsum` expression for applying the QuanTA operator.

```python
import itertools
import opt_einsum as oe

def quanta_apply_einsum_expr(N):
    current_symbols_inds = list(range(N))

    expr = "..."
    for i in current_symbols_inds:
        expr += oe.get_symbol(i)

    for (dim1, dim2) in itertools.combinations(range(-1, -N-1, -1), 2):
        symbol_ind1 = current_symbols_inds[dim1]
        symbol_ind2 = current_symbols_inds[dim2]
        symbol_ind3 = symbol_ind1 + N
        symbol_ind4 = symbol_ind2 + N
        expr += "," + \
            oe.get_symbol(symbol_ind4) + \
            oe.get_symbol(symbol_ind3) + \
            oe.get_symbol(symbol_ind2) + \
            oe.get_symbol(symbol_ind1)
        current_symbols_inds[dim1] = symbol_ind3
        current_symbols_inds[dim2] = symbol_ind4

    expr += "->..."
    for i in current_symbols_inds:
        expr += oe.get_symbol(i)

    return expr
```

Then, applying the QuanTA operator to the hidden vector is as simple as

```python
y = torch.einsum(quanta_apply_expr, x, *T)
```

Similarly, it is easy to generate the `einsum` expression for obtaining the full QuanTA operator as

```python
import itertools
import opt_einsum as oe

def quanta_op_einsum_expr(N):
    current_symbols_inds = list(range(N))

    expr = "..."
    for i in current_symbols_inds:
        expr += oe.get_symbol(i)
```

```
    for (dim1, dim2) in itertools.combinations(range(-1, -N-1, -1), 2):
        symbol_ind1 = current_symbols_inds[dim1]
        symbol_ind2 = current_symbols_inds[dim2]
        symbol_ind3 = symbol_ind1 + N
        symbol_ind4 = symbol_ind2 + N
        expr += "," + \
            oe.get_symbol(symbol_ind4) + \
            oe.get_symbol(symbol_ind3) + \
            oe.get_symbol(symbol_ind2) + \
            oe.get_symbol(symbol_ind1)
        current_symbols_inds[dim1] = symbol_ind3
        current_symbols_inds[dim2] = symbol_ind4

    expr += "->..."
    for i in current_symbols_inds:
        expr += oe.get_symbol(i)

    return expr[1:]
```

and obtaining the full QuanTA operator is

```
full_T = torch.einsum(quanta_op_expr, *T)
```

We note that the padding and truncation operators are omitted when the QuanTA operator has a different size than the original weight matrix. In addition, in our actual implementation, we use `opt_einsum` library to optimize the contraction order, reducing the contraction cost.

## H   Example Model Outputs

In this section, we provide some example output of QuanTA fine-tuned LLaMA model.

| Task | Model Output |
|------|--------------|
| **DROP** | **Prompt:** |

Passage:  Hoping to rebound from their embarrassing home loss to
↪    the Lions, the Raiders flew to Invesco Field at Mile High for
↪    an AFC West duel with the Denver Broncos.  In the first
↪    quarter, Oakland trailed early as Broncos QB Jay Cutler
↪    completed a 9-yard TD pass to WR Brandon Stokley for the only
↪    score of the period.  In the second quarter, the Raiders got
↪    on the board with kicker Sebastian Janikowski getting a
↪    38-yard field goal.  However, Denver continued to pound away
↪    as RB Cecil Sapp got a 4-yard TD run, while kicker Jason Elam
↪    got a 23-yard field goal. In the third quarter, Oakland began
↪    to come back as QB Josh McCown (who was a gametime decision
↪    heading into the game) completed a 46-yard TD pass to WR Jerry
↪    Porter for the only score of the period.  In the fourth
↪    quarter, the Raiders took the lead as DT Gerard Warren sacked
↪    Cutler in the end zone for a safety, while LB Thomas Howard
↪    returned an interception 44 yards for a touchdown (followed by
↪    a successful two-point conversion pass from McCown to WR
↪    Ronald Curry).  However, the Broncos tied the game up with
↪    Elam's 20-yard field goal.  In overtime, Oakland managed to
↪    make Denver go three-and-out on their first possession.  A
↪    33-yard run by RB LaMont Jordan helped set up Janikowski for a
↪    game-winning 52-yard field goal. Broncos head coach Mike
↪    Shanahan called timeout before the kick could begin.
↪    Janikowski's second try hit off the very tip of the left goal
↪    post and was no good, giving Denver a chance to win the game.
↪    The Broncos won with Elam getting a 23-yard field goal. With
↪    the loss, not only did the Raiders fall to 0-2, but they had
↪    lost 11-straight games (currently the NFL's longest losing
↪    streak) dating back to Week 9 of the 2006 season.
Question: How many field goals did each kicker score in the first
↪    half?
Answer:

**Output:**

1

Table H.8: Examples of QuanTA trained LLaMA2-7B Outputs for the DROP dataset.

| Task | Model Output |
|---|---|
| **BoolQ** | **Prompt:**

Please answer the following question with true or false, question:
↪ is ford escape a 4 wheel drive vehicle? Answer format:
↪ true/false

**Highest probability choice:**

Answer: the correct answer is true. |
| **SIQA** | **Prompt:**

Please choose the correct answer to the question: Carson took
↪ Lee's risk by going skydiving with him off of the plane. What
↪ will Lee want to do after? Answer1: hug Carson Answer2: buy a
↪ ticket Answer3: kick Carson. Answer format:
↪ answer1/answer2/answer3

**Highest probability choice:**

Answer: the correct answer is answer1. |
| **SIQA** | **Prompt:**

Please choose the correct ending to complete the given sentence:
↪ Personal Care and Style: [header] How to make ice balls
↪ [title] Buy a package of water balloons. [step] This method is
↪ cheap, quick, and easy-perfect if you don't want to spend
↪ money on specialty molds for making ice balls. All you'll need
↪ is a few round water balloons (and, of course, water and a
↪ freezer. Ending1: ) [substeps] Uninflated balloons: this
↪ method requires 2 balls, 1 ice cream stick and 2 water
↪ balloons in a large bag (1 at a time). Open the sides of your
↪ volcano and shake the tupperware from side to side a few times.
↪ Ending2: ) [substeps] Of course, there is no telling if the
↪ water balloons will be successful; you just know that they
↪ will eventually explode on the first try. Plus, they're
↪ basically too small to correctly hang from your roof. Ending3:
↪ ) [substeps] Remember that, if you're using your ice balls to
↪ cool your drinks, they'll need to be able to fit inside your
↪ glasses. Thus, when it comes to water balloons, generally, the
↪ smaller, the better. Ending4: ) [substeps] Use unopened water
↪ balloons, because they tend to be too full. Plastic ones are
↪ dry and will be harder to put in the freezer. Answer format:
↪ ending1/ending2/ending3/ending4

**Highest probability choice:**

Answer: the correct answer is ending3. |

Table H.9: Examples of QuanTA trained LLaMA3-8B Outputs for selected commonsense datasets.

| Task | Model Output |
|------|--------------|
| MAWPS | **Prompt:** |

```
Bob will rent a car for the weekend . He can choose one of 2
↪   payment plans . The first plan costs 57.98 for 2 days plus 13
↪   cents per mile . The second plan costs 45.98 for 2 days plus
↪   18 cents per mile . How many miles does Bob need to drive for
↪   the 2 plans to cost the same ?
Answer:
```

**Output:**

```
Let's assume that Bob drives x miles.  For the first plan, the
↪   cost would be: 57.98 + 0.13x  For the second plan, the cost
↪   would be: 45.98 + 0.18x  We want to find the value of x such
↪   that the two costs are equal.  So, we can set up an equation:
↪   57.98 + 0.13x = 45.98 + 0.18x  Simplifying this equation, we
↪   get:  0.05x = 12.00  Dividing both sides by 0.05, we get:  x =
↪   240  Therefore, Bob needs to drive 240 miles for the two plans
↪   to cost the same.
```

Table H.10: Examples of QuanTA trained LLaMA2-7B Outputs for a selected arithmetic dataset.

