# OpenReview forum: "QuanTA: Efficient High-Rank Fine-Tuning of LLMs with Quantum-Informed Tensor Adaptation"
_NeurIPS.cc/2024/Conference — NeurIPS 2024 poster_

### Official Review · Reviewer_H46K · 2024-06-15

**Soundness:** 4
**Presentation:** 3
**Contribution:** 3
**Rating:** 7
**Confidence:** 3

**Summary:**

This paper proposes a novel PEFT method that inspired by the Quantum Circuit. Their method is theoretically supported by the universality theorem and the rank representation theorem to achieve efficient high-rank adaptations on various downstream tasks. The QuanTA method surpasses LoRA, DoRA and even fine-tuning in some cases.

**Strengths:**

1. The paper is well-motivated and presents sufficient theoretical prove and justifications.
2. The experiments are abundant and the model performs very well compared with other baselines with much less activated parameters
3. A well-written paper

**Weaknesses:**

1. I'm curious about the performance of QuanTA on textual understanding (classification) tasks with BERT or RoBERTa. Afterall, all these PEFT models were initially applied to text classification tasks (LoRA, Adapter...)
2. Do you have experiments that combine other PEFT methods with your QuanTA? Since the number of activated parameters of your method is much less than baselines. Can we further boost the model performance by adding your method to others?

**Questions:**

Please refer to the weaknesses section.

**Limitations:**

Yes

---

> ### Author Rebuttal · Authors · 2024-08-07
>
> We are very appreciative of the reviewer's valuable comments and suggestions! We address the raised weaknesses below.
>
> **Weaknesses:**
>
> >I'm curious about the performance of QuanTA on textual understanding (classification) tasks with BERT or RoBERTa. Afterall, all these PEFT models were initially applied to text classification tasks (LoRA, Adapter...)
>
> Thanks for the comments! In the original manuscript, we focus on LLaMA benchmarks because we believe such tasks are usually more challenging than tasks with BERT or RoBERTa. We note that one of the key questions that QuanTA addresses is high-rank fine-tuning of challenging tasks (which are non-trivial for existing methods), so that we do not focus on the textual understanding tasks that are usually considered to be simpler.  However, it is still very valuable to examine QuanTA in the context of text classification for completeness. In the table below, we compare QuanTA and LoRA (fine-tuning RoBERTa) on five natural language understanding tasks. While we would like to include more textual understanding benchmarks, some of the benchmarks have a large training set, and we were limited by our computational resources and the rebuttal timeline to include additional experiments. We note that while the results are less impressive than our experiments on LLaMA, QuanTA still achieves similar or better results than LoRA with fewer parameters. In addition, when fine-tuning BERT or RoBERTa, the output classifier layer usually contains most of the trainable parameters, making the parameter reduction advantage of QuanTA appear to be less significant. Even so, QuanTA still achieves better results with fewer parameters.
>
>
> | PEFT Method          | # Params (%) | SST-2 | MRPC | CoLA | RTE | STS-B |
> |----------------------|--------------|-------|------|------|---------|--------|
> | LoRA                 | 0.71%        |  **94.01**  | 91.48 | **62.08** | 74.51    | 90.48   |
> | **QuanTA (Ours)**    | 0.62%       | 93.81 |**91.67**| **62.08** |**77.26**| **90.68**   |
>
>
> >Do you have experiments that combine other PEFT methods with your QuanTA? Since the number of activated parameters of your method is much less than baselines. Can we further boost the model performance by adding your method to others?
>
> Thanks for the comment and this is a very good point. While we did not have the time to explore such possibilities, it could be very advantageous to combine QuanTA with other PEFT methods. For example, it would be possible to do Q-QuanTA to further reduce the cost by quantizing the QuanTA weights. In addition, the current QuanTA weights are mostly "square". It would also be interesting to further combine QuanTA and LoRA, making low-rank parameterizations of QuanTA weights. Many other modifications or variants of LoRA could also be adapted to QuanTA to further improve our results. It will be a good future direction to further boost the model performance by integrating QuanTA and other methods.
>
> **Questions:**
>
> >Please refer to the weaknesses section.
>
> >**Rating: 7:** Accept: Technically solid paper, with high impact on at least one sub-area, or moderate-to-high impact on more than one areas, with good-to-excellent evaluation, resources, reproducibility, and no unaddressed ethical considerations.
>
>
> We are very grateful for the reviewer's comments and questions for improving our work. We have supported new benchmarks and discussions to address the concerns. It would be greatly appreciated if the reviewer could consider raising the score.

---

> > ### Author Response · Authors · 2024-08-11
> >
> > We are really grateful for the reviewer's suggestions and comments. We have included new discussions and experiments in the rebuttal, and it will be appreciated to know if there is any further suggestions!

---

> > > ### Comment · Reviewer_H46K · 2024-08-13
> > >
> > > Thank you for your responses and further experiments. I'll keep my score and tend to accept the paper.

---

### Official Review · Reviewer_Qk3g · 2024-07-10

**Soundness:** 3
**Presentation:** 3
**Contribution:** 3
**Rating:** 8
**Confidence:** 3

**Summary:**

The paper proposes an efficient fine tuning method inspired from quantum circuits for large scale pretrained models. This is an alternative lorank approximation to weight updates (LoRA). The paper considers decomposing the dimension d of the square weight matrix as  d=d1xd2x...xdN) .The hidden vector in d dimensions can then be considered as a vector in the corresponding tensor product space of qudits. The action of the weight update is then considered as a sequence of quantum gates applied to these qudits.
The weighted layer is then defined as y = W_\theta x = W_0 x + T_\theta x - Sx where W_0 is the initial weight matrix, T_\theta is trainable quantum circuit and S is fixed to be initial circuit for T_\theta.
Through experiments on multiple datasets they are able to beat other fine tuning methods considering versions of Llama as the base model.
The paper also discusses the theoretical results to show the universality of tensor decomposition into tensors acting on atmost 2 axis. They have also derived lower and upper bounds on the rank of the overall tensor product operator in terms of ranks of individual tensors.
The work resolves the issue with low rank approximation which may not always be sufficient as shown through experiments.

**Strengths:**

1. The idea of using tensor decomposition for PEFT is original , novel and would find applications in the area of LLM fine tuning.
2. The experimental results show promising results on multiple datasets including DROP, CommonSense Reasoning and Arithmetic Reasoning.
3. The paper is backed by theoretical justification for universality of tensor product decomposition.
4. The experiments also justify the need for higher rank approximations.

**Weaknesses:**

1. The paper does not discuss the ordering over the pairs of qubits chosen. Also in general the universality theorem for tensor product might involve picking a set of pairs of qubits where a given pair might participate multiple times in the circuit. Is picking a pair of qubits only once always sufficient?

2.  A minor concern could also be that the dimension of of hidden state needs to be factorizable as product of small dimensions. This would typically not be a concern in practice since these are powers of 2, so would work perfectly in those scenarios.

**Questions:**

1. 1. The paper does not discuss the ordering over the pairs of qubits chosen. Also in general the universality theorem for tensor product might involve picking a set of pairs of qubits where a given pair might participate multiple times in the circuit. Is picking a pair of qubits only once always sufficient?

**Limitations:**

The authors have addressed their limitations and societal impact  well.

---

> ### Author Rebuttal · Authors · 2024-08-07
>
> We are very appreciative of the reviewer's valuable comments and suggestions! We address the raised weaknesses and questions below.
>
> **Weaknesses:**
>
> >The paper does not discuss the ordering over the pairs of qubits chosen.
>
> Thanks for the question. While the ordering of pairs in general should not significantly affect the results, the specific choices in this paper is generated with the following code:
> ```
> itertools.combinations(range(-1, -N-1, -1), 2)
> ```
> with N being the number of axes. The details of the circuit architecture, including the ordering of pairs are already described in Appendix E1 and F. We update the paper to further improve the clarity of this point.
>
> >Also in general the universality theorem for tensor product might involve picking a set of pairs of qubits where a given pair might participate multiple times in the circuit. Is picking a pair of qubits only once always sufficient?
>
> The reviewer is absolutely correct. Picking each pair of qudits only once does not necessarily satisfy the universality theorem. The universality theorem can be paraphrased as the following: given sufficient layers, QuanTA has the same representation as full fine-tuning. In practical cases, full fine-tuning is often not needed, either because the subsequent tasks are close to the pretraining data, or due to the small sample sizes of the subsequent tasks. Therefore, it is usually not necessary to have multiple gates for each pair of qudits. In addition, even with a single gate for each pair of qudits, QuanTA can already parameterize full-rank matrices and introduce correlations to any pair of qudits, which is not possible for existing methods such as LoRA or alike. While we find, empirically, that a single gate for each pair of qudits is sufficient for all the fine-tuning tasks we tried, additional gates can be employed when the task is significantly harder. This flexibility gives an additional tuning knob for the expressivity. Exploring how the performance scales as the number of gates on each pair of axes is also a very good theoretical question that involves further understanding of quantum information theory and will be a nice direction for future exploration.
>
> >A minor concern could also be that the dimension of of hidden state needs to be factorizable as product of small dimensions. This would typically not be a concern in practice since these are powers of 2, so would work perfectly in those scenarios.
>
> Thanks for the comment. Yes, as the reviewer has noted, many existing LLMs have the hidden state chosen as a power of 2. However, we would like to note that, while the optimal design is not clear, QuanTA is flexible to be incorporated in LLMs with unfactorizable hidden dimensions. As already described in Appendix B, it is completely valid to either pad or truncate the hidden state to the nearest factorizable size in any layer before applying QuanTA. This also works in the inference stage, as the QuanTA matrices can also be padded or truncated analogously to merge with the original weight matrices to avoid any inference latency. We improve Appendix B to make this point clearer in the new version of the paper.
>
> **Questions:**
> >The paper does not discuss the ordering over the pairs of qubits chosen. Also in general the universality theorem for tensor product might involve picking a set of pairs of qubits where a given pair might participate multiple times in the circuit. Is picking a pair of qubits only once always sufficient?
>
> Thanks for the reviewer's question! We provided the answers above in the weaknesses section.
>
> >**Rating: 8:** Strong Accept: Technically strong paper, with novel ideas, excellent impact on at least one area, or high-to-excellent impact on multiple areas, with excellent evaluation, resources, and reproducibility, and no unaddressed ethical considerations.
>
> We are very grateful for the reviewer's comments and questions for improving our work. We have supported a new discussion to address the concerns. It would be deeply appreciated if the reviewer could consider raising the score.

---

> ### Author Response · Authors · 2024-08-11
>
> We are really grateful for the reviewer's suggestions and comments. We have included new discussions and experiments in the rebuttal, and it will be appreciated to know if there is any further suggestions!

---

> > ### Comment · Reviewer_Qk3g · 2024-08-12
> >
> > Thanks for answering the questions. I acknowledge that I have read the rebuttal.

---

### Official Review · Reviewer_GYap · 2024-07-11

**Soundness:** 4
**Presentation:** 3
**Contribution:** 4
**Rating:** 8
**Confidence:** 3

**Summary:**

The authors address the issue that low-rank adaptation methods fail when applied to more complex tasks. They clearly present this motivation through experiments on two datasets of varying complexity. Drawing inspiration from quantum information science, the authors propose using the formalism from quantum circuits, which describe operations on quantum systems whose dimensions increase exponentially with the number of qubits.

**Strengths:**

1. The paper is clearly motivated and well-written, with the only challenging aspect being the tensor formalism. However, the authors acknowledge this difficulty and frequently provide specific examples to aid understanding.
2. Integrating quantum concepts in fine-tuning appears to be a novel approach and demonstrates promising results.

**Weaknesses:**

1. Please clarify the start and end limits in the summations.
2. The authors justify their results by stating that adaptive methods based on low-rank decomposition perform poorly on more complex tasks. Why is LoRA the only baseline method considered? Why did the authors not compare their method with high-rank methods, such as MoRA or KronA?
3. I appreciate that the authors devoted some attention to the complexity of the proposed methods, but it would be very beneficial if they also provided some evaluation of training time and GPU memory usage in comparison to other baselines.

**Questions:**

1. From a practical point of view, is there any preference regarding the decomposition of the output dimension? For example, should we favour smaller-dimensional axes? There are many possibilities for such a decomposition, so any tips that would help in exploring this hyperparameter subspace would be valuable.
2. Why is the method limited to only two-axis gates? Multi-axis gates should be more expressive, correct? In quantum computing, we typically restrict ourselves to two-qubit gates due to the difficulty of implementing multi-qubit gates on real devices. However, this constraint does not apply here.
3. I understand the motivation of this work, where the authors show the limitations of low-rank adaptation methods in the context of more complex reasoning tasks. Therefore, will the proposed method provide any benefits in terms of fine-tuning in the domain of computer vision?
4. How large is the $\alpha$ parameter? My understanding is that $\alpha$ dictates the number of gates used in the circuit, and subsequently, the expressiveness of the final circuit is influenced by this value and the arrangement of these gates. Is this correct? If so, what is the relationship between the performance of the proposed method and the value of $\alpha$?

**Limitations:**

yes

---

> ### Author Rebuttal · Authors · 2024-08-07
>
> We are deeply appreciative of the reviewer's valuable comments and suggestions! We address the raised weaknesses and questions below. Due to the rebuttal length limitation, we address the weaknesses in this post and the questions in a new post.
>
> **Weaknesses:**
>
> >Please clarify the start and end limits in the summations.
>
> Thanks a lot for the suggestion! For $i_m$, $j_m$ or $k_m$, the summation is from 1 to $d_m$, the number of dimensions of the $m$-th axis. For $\alpha$, the summation (or product) is from $1$ to the number of matrices. We update the manuscript and include the limits to improve clarity.
>
> >The authors justify their results by stating that adaptive methods based on low-rank decomposition perform poorly on more complex tasks. Why is LoRA the only baseline method considered? Why did the authors not compare their method with high-rank methods, such as MoRA or KronA?
>
> Thanks for the suggestion and we fully agree that it would be beneficial to include comparisons with other high-rank methods. We would like to note that MoRA is concurrent to our work and was posted on arXiv two days before the NeurIPS submission deadline, making it very difficult to include comparisons at the time of submission. In the tables below, we provide additional benchmarks with MoRA and KronA to further provide evidence of the effectiveness of QuanTA.
>
> Comparison of fine-tuning LLaMA2 7 billion model on DROP dataset using QuanTA, LoRA, MoRA, and KronA:
>
> | PEFT Method                      | # Params (%) | F1 Score (↑) |
> |----------------------------------|--------------|--------------|
> | LoRA_r=8                         | 0.062%       | 54.0         |
> | LoRA_r=32                        | 0.249%       | 54.8         |
> | LoRA_r=128                       | 0.996%       | 56.2         |
> | MoRA_r=8                         | 0.062%       | 58.6         |
> | MoRA_r=32                        | 0.249%       | 58.2         |
> | MoRA_r=128                       | 0.996%       | 58.9         |
> | KronA_64-64                      | 0.008%       | 50.9         |
> | KronA_256-16                     | 0.062%       | 57.7         |
> | KronA_1024-4                     | 0.996%       | 58.5         |
> | **QuanTA_16-8-8-4 (Ours)**       | 0.041%       | 59.5         |
> | **QuanTA_16-16-16 (Ours)**       | 0.261%       | **59.6**     |
>
> Comparison of fine-tuning LLaMA3 8 billion model on various commonsense tasks using QuanTA, LoRA, KronA:
>
> | PEFT Method          | # Params (%) | BoolQ | PIQA | SIQA | HellaS. | WinoG. | ARC-e | ARC-c | OBQA | Avg.  |
> |----------------------|--------------|-------|------|------|---------|--------|-------|-------|------|-------|
> | LoRA                 | 0.70%        |  70.8  | 85.2 | 79.9 | 91.7    | 84.3   | 84.2  | 71.2  | 79.0 | 80.8  |
> | KronA                | 0.052%       |  72.9  | 87.1 | 80.6 | 92.1    | 85.1   | 87.8  | 76.0  | 84.3 | 83.2  |
> | **QuanTA (Ours)**    | 0.035%       |  **74.3**  |**88.1**| **81.8** |  **95.1** | **87.3**   | **91.1**  | **81.7**  | **87.2** | **85.8** |
>
>
> For the commonsense tasks, while we would like to compare QuanTA to both KronA and MoRA, we find MoRA can be sensitive to hyperparameters. Due to the limited computational resources and the large training set of COMMONSENSE170K, we were unable to find a good combination of hyperparameters for MoRA during the rebuttal period. Therefore we only include MoRA benchmarks on the DROP dataset. The new results demonstrate that while MoRA and KronA can improve significantly from base LoRA, QuanTA still achieves better performance while using fewer parameters. It further demonstrates QuanTA's effectiveness considering the concurrent development of MoRA.
>
> We would like to further note that while KronA can parameterize full-rank matrices using the Kronecker product, it is still "low-rank" from a different perspective: in quantum circuit language, KronA parameterizes only single-qudit gates, which does not introduce entanglements (or correlations between qudits). Introducing correlations with two-qudit gates is crucial for universal representation, which is lacking in KronA. Our QuanTA method is designed systematically with a theoretical guarantee of its universality, rank representation, and composition openness (Theorem I to III).
>
> >I appreciate that the authors devoted some attention to the complexity of the proposed methods, but it would be very beneficial if they also provided some evaluation of training time and GPU memory usage in comparison to other baselines.
>
> Thanks for the suggestion! To better understand the time and space complexity of our method, we run additional experiments and profile the average runtime per step and the maximum allocated GPU memory for training LLaMA 3 8 billion model on the COMMONSENSE170K dataset with a batch size of 4 (to fully saturates the GPU memory). The result is shown below.
>
> | PEFT Method          | Average Runtime per Step (second) | Maximum Allocated GPU Memory (GB) |
> |----------------------|------|-------|
> | LoRA                 | 0.76 | 76.8  |
> | MoRA                 | 0.79 | 78.5  |
> | KronA                | 0.73 | 76.4  |
> | **QuanTA (Ours)**    | 0.75 | 75.6  |
>
> We find that while all methods have very similar runtime and memory allocation--potentially because most of the resources are devoted to running the base model--our method may still have a slight advantage. All the experiments are performed on A100 GPUs with 80GB VRAM. We would like to further note that the current implementation of QuanTA is based on a single ```einsum``` operation. Further optimizations using custom CUDA kernels could be possible.
>
> Due to the rebuttal length limitation, we address the questions in the next post.

---

> ### Author Response · Authors · 2024-08-07
> **Additional Rebuttal due to Length Limitation**
>
> We address the questions in this post.
>
> **Questions:**
>
> >From a practical point of view, is there any preference regarding the decomposition of the output dimension? For example, should we favour smaller-dimensional axes? There are many possibilities for such a decomposition, so any tips that would help in exploring this hyperparameter subspace would be valuable.
>
> Thanks for the question! While our paper already includes experiments using two different decompositions in Table 2, we did not systematically explore how the decomposition affects the performance. Nevertheless, we empirically found that it is usually sufficient to choose either 3 or 4 axes, with a maximum of 16 dimensions per axes for the LLaMA architecture. We update our paper with this comment to guide readers on how to choose these hyperparameters. The choice of such decomposition could also be related to the understanding of entanglement in quantum information theory. It will be interesting to further explore additional decompositions from both theoretical and practical perspectives to improve our results in future work.
>
> >Why is the method limited to only two-axis gates? Multi-axis gates should be more expressive, correct? In quantum computing, we typically restrict ourselves to two-qubit gates due to the difficulty of implementing multi-qubit gates on real devices. However, this constraint does not apply here.
>
> Thanks for the comment. Indeed, multi-axis gates are also possible, and there is no limitation of such gates in our implementation. While such gates can be more expressive than two-axis gates, they also contain more parameters, so the additional benefits are unclear. In addition, it is possible to group together multiple axes, and rewrite multi-axis gates as two-axis gates. Given that we have proved the universality theorem of two-axis gates for QuanTA, we find it parameter efficient and effective to use the two-axis gate and hence have not explored additional multi-axis gates. However, such exploration on the multiple-axis gate will be an interesting direction for future study, and we update our paper with additional comments on multi-axis gates.
>
> >I understand the motivation of this work, where the authors show the limitations of low-rank adaptation methods in the context of more complex reasoning tasks. Therefore, will the proposed method provide any benefits in terms of fine-tuning in the domain of computer vision?
>
> Thanks for the comment. It would be very interesting to explore how QuanTA could improve the state-of-the-art in the domain of computer vision. One important feature of QuanTA is for high-rank fine-tuning of complicated tasks. It will be an important direction to apply QuanTA to challenging computer vision tasks and probably even to video generation in future work.
>
> >How large is the $\alpha$ parameter? My understanding is that $\alpha$ dictates the number of gates used in the circuit, and subsequently, the expressiveness of the final circuit is influenced by this value and the arrangement of these gates. Is this correct? If so, what is the relationship between the performance of the proposed method and the value of $\alpha$?
>
> Thanks for the question! The reviewer's understanding is correct. $\alpha$ is the index of the gates used in the parameterization. In this work, we choose to apply one gate for each pair of axes, and therefore there are 3 gates in total for the case of 3 axes, and 6 gates in total for the case of 4 axes. In Appendix E1, we already described the exact circuit architecture used and provided examples of up to 5 axes. We further update our paper to make the range of $\alpha$ more clear in the new version. The exact relation between the performance of the proposed method and the value of $\alpha$ is also a very good theoretical question that may have additional connections to the understanding of quantum information theory, which will be a nice direction for future exploration.
>
>
> >**Rating:** 7: Accept: Technically solid paper, with high impact on at least one sub-area, or moderate-to-high impact on more than one areas, with good-to-excellent evaluation, resources, reproducibility, and no unaddressed ethical considerations.
>
>
> We are deeply grateful for the reviewer's comments and questions for improving our work. We have supported new benchmarks and discussions to address the concerns. It would be appreciated if the reviewer could consider raising the score.

---

> > ### Comment · Reviewer_GYap · 2024-08-10
> >
> > Thank you for comprehensively answering my questions and adding new pools and time measurements. I raised my rating.

---

> > > ### Author Response · Authors · 2024-08-11
> > >
> > > We sincerely appreciate the reviewer's feedback and support!

---

### Official Review · Reviewer_ddRu · 2024-07-12

**Soundness:** 3
**Presentation:** 3
**Contribution:** 2
**Rating:** 6
**Confidence:** 3

**Summary:**

This work proposed a method, QuanTA, that leverages quantum-inspired techniques for efficient fine-tuning of large pre-trained language models. The authors show that it outperforms Low-Rank Adaptation (LoRA) in complex tasks and has significant improvements in common-sense and arithmetic reasoning.

**Strengths:**

A novel idea to represent the weight updating by a quantum circuit inspired tensor adaptation.

**Weaknesses:**

the comparison of related tensor-based approaches is missing, it is unclear the outperformance of the proposed method over related previous works.

**Questions:**

1. As quantum circuit can also be written in the form of a tensor network, what are the main advantages of the proposed method compared to other previous tensor-based PEFT？ such as FACT(https://arxiv.org/pdf/2212.03145) or KronA(https://arxiv.org/pdf/2212.10650)
2. As the proposed method employs a quantum circuit to replace the low-rank adaptation,  there are various circuit architectures, which kind of circuit ansatz (architecture) is used here, arbitrary or specific? and what is the motivation for choosing such one?

**Limitations:**

no negative societal impact of their work

---

> ### Author Rebuttal · Authors · 2024-08-07
>
> We thank the reviewer for the valuable comments and suggestions! We address the raised weaknesses and questions below.
>
> **Weaknesses:**
>
> >the comparison of related tensor-based approaches is missing, it is unclear the outperformance of the proposed method over related previous works.
>
> We thank the reviewers for the suggestion of comparing QuanTA to other tensor-based approaches. Here, we run additional experiments on tensor-based methods. In particular, we compared our method with KronA(https://arxiv.org/pdf/2212.10650) and LoRETTA(https://arxiv.org/pdf/2402.11417, another tensor-based method generalizing LoRA). Although we would like to include comparisons with FacT, we weren't able to perform the experiment due to the lack of open-source implementation.
>
> Comparison of fine-tuning LLaMA2 7 billion models on DROP dataset using QuanTA, LoRA, KronA, and LoRETTA:
>
> | Method | Params (%) | F1 (↑) |
> |----|----|-----|
> | LoRA_r=8 | 0.062% | 54.0 |
> | LoRA_r=32 | 0.249% | 54.8 |
> | LoRA_r=128 | 0.996% | 56.2 |
> | KronA_64-64 | 0.008% | 50.9 |
> | KronA_256-16 | 0.062% | 57.7 |
> | KronA_1024-4 | 0.996% | 58.5 |
> | LoRETTA_r=8 | 0.009% | 48.6 |
> | LoRETTA_r=32 | 0.083% | 54.9 |
> | LoRETTA_r=128 | 1.254% | 59.1 |
> | **QuanTA_16-8-8-4** | 0.041% | 59.5 |
> | **QuanTA_16-16-16** | 0.261% | **59.6** |
>
> Comparison of fine-tuning LLaMA3 8 billion model on various commonsense tasks using QuanTA, LoRA, KronA, and LoRETTA:
>
> | Method | Params (%) | BoolQ | PIQA | SIQA | HellaS. | WinoG. | ARC-e | ARC-c | OBQA | Avg. |
> |--------|------------|-------|------|------|---------|--------|-------|-------|------|------|
> | LoRA   | 0.70%      | 70.8  | 85.2 | 79.9 | 91.7    | 84.3   | 84.2  | 71.2  | 79.0 | 80.8 |
> | KronA  | 0.052%     | 72.9  | 87.1 | 80.6 | 92.1    | 85.1   | 87.8  | 76.0  | 84.3 | 83.2 |
> | LoRETTA| 0.13%      |**74.3**| 87.5 | 80.9 | 94.5    | 86.7   |**92.1**| 81.5  | 85.8 | 85.4 |
> | **QuanTA** | 0.035% |**74.3**|**88.1**|**81.8**|**95.1**|**87.3**| 91.1 |**81.7**|**87.2**|**85.8**|
>
> From the results above, we find that while these tensor-based approaches could be better than LoRA, they are not as good as our QuanTA method, in terms of both performance and parameter efficiency.
>
> **Questions:**
>
> >As quantum circuit can also be written in the form of a tensor network, what are the main advantages of the proposed method compared to other previous tensor-based PEFT？ such as FACT(https://arxiv.org/pdf/2212.03145) or KronA(https://arxiv.org/pdf/2212.10650)
>
> Thanks for the question. This is a very good point. While a quantum circuit can be written as a form of tensor network, not all tensor networks have the same expressivity. Many existing tensor-based approaches still face low-rank limitations in certain ways. For example, FacT generalizes LoRA by viewing the weights over all layers jointly as a tensor to perform tensor decomposition. However, the proposed tensor methods (FacT-TT and Fact-TK) do not resolve the low-rank limitation. KronA, on the other hand, can parameterize full-rank matrices using the Kronecker product. However, it is still "low-rank" from a different perspective: in quantum circuit language, KronA parameterizes two single-qudit gates, which cannot introduce entanglements (correlations) between qudits. Introducing correlations with two-qudit gates is crucial for universal representation, which is lacking in KronA. Our QuanTA method is designed in a systematic way with a theoretical guarantee of its universality, rank representation, and composition openness (Theorem I to III). We also include the new benchmarks for KronA and LoRETTA above and demonstrate QuanTA achieves superior performance compared to other tensor-based methods. FacT experiment is omitted due to the previously mentioned reason.
>
> >As the proposed method employs a quantum circuit to replace the low-rank adaptation, there are various circuit architectures, which kind of circuit ansatz (architecture) is used here, arbitrary or specific? and what is the motivation for choosing such one?
>
> Thanks for raising a very good point. In conventional quantum circuits (or quantum machine learning), people usually fix the two-qubit gates and only allow arbitrary single-qubit rotations. Different circuit architectures usually differ in the choice of two-qubit gates (CNOT or CZ) and the parameterization of single-qubit gates (X/Y/Z rotations). The architecture used in our work can be viewed as a generalization of conventional architectures in three ways: (1) we generalize unitary matrices to arbitrary matrices because classical computation does not have the unitarity constraint; (2) we generalize the circuit from qubits to "qu*d*its", meaning each dimension is generalized from "2" to arbitrary numbers; and (3) we allow the two-qudit gates to be fully parameterizable, instead of taking fixed forms. Because of the third generalization, the removal of single-qudit gates does not affect the expressivity, as they can be *merged* into the fully parameterizable two-qudit gates. Using this generalized architecture, we apply one two-qudit gate for each pair of qudits (axes or dimensions). This is one of the innovative designs of QuanTA that allows both full-rank matrix parameterization and entanglement generation between different qudits, while existing tensor-based methods fail to do so. In Appendix E1, we already described the architecture and gave examples for up to 5 qudits. Thanks for the reviewer's question, we further update this section to improve the clarity of the circuit architecture in the new version of our paper.
>
> >**Rating:** 5: Borderline accept: Technically solid paper where reasons to accept outweigh reasons to reject, e.g., limited evaluation. Please use sparingly.
>
> We are very grateful for the reviewer's valuable comments and questions for improving our work. We have supported new benchmarks and discussions to address the concerns. We would be very appreciative if the reviewer could consider raising the score.

---

> > ### Comment · Reviewer_ddRu · 2024-08-13
> >
> > Thank you for the response and most of my questions are solved. I would happy to raise my score.

---

> ### Author Response · Authors · 2024-08-11
>
> We are really grateful for the reviewer's suggestions and comments. We have included new discussions and experiments and received positive feedback from other reviewer. It will be appreciated to know if there is any further suggestions!

---

### Author Rebuttal · Authors · 2024-08-07

We thank all the reviewers for their valuable comments! We appreciate all reviewers for noting the novelty of our work as well as our strong theory and experiment results.

Below, we summarize the major concerns of the reviewers and our responses.

>In general, reviewers agree that our results are already impressive, although some reviewers are interested in seeing additional benchmarks: (Reviewer ddRU) QuanTA vs other tensor-based fine-tuning methods; (Reviewer GYap) QuanTA vs other high-rank fine-tuning methods; and (Reviewer H46K) QuanTA's performance on textual understanding tasks.

To address the reviewer's concern, we perform new experiments comparing QuanTA with KronA (tensor-based and high-rank), LoRETTA (tensor-based), and MoRA (high-rank) and show that although these methods can achieve better performance than LoRA, they are not as good as QuanTA in terms of both performance and parameter efficiency. In addition, we compare QuanTA and LoRA on text understanding tasks by fine-tuning the RoBERTA base model. We show that QuanTA is still advantageous in this application.

>Reviewer GYap acknowledges our theoretical discussions on the space and time complexity of QuanTA, but is interested in the actual runtime and memory cost for the experiments.

To answer this question, we include new results profiling the runtime per iteration as well as the memory usage during training.

>While reviewers in general agree that our paper is clear and well-written, especially considering the complexity of tensor operations, reviewers have some questions requiring further clarifications of certain parts of our paper.

In our response below, we discuss all the points in detail and we update the new version of our paper to further clarify these points.

>Reviewers are also interested in further explorations of QuanTA, such as additional experimentations of circuit architecture, QuanTA's performance in computer vision tasks, and combining other PEFT methods with QuanTA.

We thank all the reviewers for these great suggestions and believe it would be important to explore them in subsequent works.

We would like to further highlight the novelty and significance of our work: (1) as a quantum-informed algorithm, our work bridges quantum computation and machine learning, opening up more opportunities for quantum-inspired ideas or even real quantum hardware realization to improve LLMs (2) our method allows high-rank fine-tuning that is extremely parameter efficient with no inference overhead, achieving state-of-the-art results in multiple benchmarks.

---

### Author Response · Authors · 2024-08-14

We are deeply grateful for all the suggestions and discussions from the reviewers. We have included new experiments and elaborations to provide strong support for QuanTA, which has also received the reviewers’ recognition with scores increased. We hope that our novel work QuanTA will make an impactful contribution to the conference by advancing efficient high-rank fine-tuning of LLMs with innovative ideas bridging machine learning and quantum simulations.

---

### Decision · Program_Chairs · 2024-09-25

**Decision:**

Accept (poster)

**Comment:**

The paper presents a novel fine-tuning method inspired by quantum circuit-like reparameterization, which is clearly motivated and well-written. The authors have done a commendable job addressing the inherent complexity of tensor formalism, offering specific examples that enhance the reader's understanding. The integration of quantum concepts into the fine-tuning process appears to be an innovative approach, yielding promising results that contribute positively to the field.

There are several key areas that require attention for the camera-ready version. It is crucial to include all the additional results provided during the rebuttal phase, including the additional baselines, runtime analysis, and the various discussion points. These elements are essential for a comprehensive evaluation of the proposed method. Additionally, the discussion on the "composition openness" of QuanTA compared to LoRA is somewhat misleading, as LoRA adapters are not composed directly with each other but with a nonlinear operator in between. It is recommended to revise this section for clarity. Overall, with these revisions, the paper is strong and merits acceptance.